# Quantum-dot-labeled synuclein seed assay identifies drugs modulating the experimental prion-like transmission

Yukio Imamura[1], Ayami Okuzumi[1,2], Saki Yoshinaga[1], Akiko Hiyama[1], Yoshiaki Furukawa [3], Tomohiro Miyasaka[4], Nobutaka Hattori[2] & Nobuyuki Nukina [1✉]

Synucleinopathies are neurodegenerative disorders including Parkinson disease (PD), dementia with Lewy body (DLB), and multiple system atrophy (MSA) that involve deposits of the protein alpha-synuclein (α-syn) in the brain. The inoculation of α-syn aggregates derived from synucleinopathy or preformed fibrils (PFF) formed in vitro induces misfolding and deposition of endogenous α-syn. This is referred to as prion-like transmission, and the mechanism is still unknown. In this study, we label α-syn PFF with quantum dots and visualize their movement directly in acute slices of brain tissue inoculated with α-syn PFF seeds. Using this system, we find that the trafficking of α-syn seeds is dependent on fast axonal transport and the seed spreading is dependent on endocytosis and neuronal activity. We also observe pharmacological effects on α-syn seed spreading; clinically available drugs including riluzole are effective in reducing the spread of α-syn seeds and this effect is also observed in vivo. Our quantum-dot-labeled α-syn seed assay system combined with in vivo transmission experiment reveals an early phase of transmission, in which uptake and spreading of seeds occur depending on neuronal activity, and a later phase, in which seeds induce the propagation of endogenous misfolded α-syn.

[1] Laboratory of Structural Neuropathology, Doshisha University Graduate School of Brain Science, 1-3 Miyakodanitatara, Kyotanabe-shi, Kyoto 610-0394, Japan. [2] Department of Neurology, Juntendo University Graduate School of Medicine, 2-1-1 Hongo, Bunkyo-ku, Tokyo 113-8421, Japan. [3] Department of Chemistry, Keio University, 3-14-1 Hiyoshi, Kohoku, Yokohama, Kanagawa 223-8522, Japan. [4] Department of Neuropathology, Faculty of Life and Medical Sciences, Doshisha University, 1-3 Miyakodanitatara, Kyotanabe-shi, Kyoto 610-0394, Japan. ✉email: nobuyukinukina@me.com

Parkinson disease (PD) is a movement disorder characterized by bradykinesia, rigidity, and tremor. The pathological hallmarks of PD include progressive neuronal loss in substantia nigra and Lewy bodies, which are cytoplasmic inclusions composed of abnormally aggregated α-synuclein (α-syn) protein[1–3]. α-syn is deposited in a phosphorylated form as Lewy bodies and Lewy neurites in PD and dementia with Lewy body (DLB); in multiple system atrophy (MSA), it is deposited as glial cytoplasmic inclusions (GCI) in oligodendrocytes[4]. Disorders with α-syn deposits are called synucleinopathies[5]. Although the pathological process between those α-syn depositions and neurodegeneration remains unclear, other neurodegenerative disorders also show aggregates of misfolded proteins such as amyloid-beta (Aβ) and tau in Alzheimer disease (AD) and TDP-43 in amyotrophic lateral sclerosis (ALS); these misfolded protein aggregates all form amyloid fibrils, which contain highly ordered cross-β sheet structure[6]. These common features suggest that protein misfolding and aggregation could be a primary event in neurodegeneration.

Pathological studies of the PD brain have revealed that α-syn pathology spreads stereotypically from the olfactory bulb and the gut to the brain stem, and it has been postulated that the spreading occurs in connected brain regions over the course of several years[7,8]. Tau pathology in AD also shows stereotypical spreading along neuroanatomical connections[9,10]. Recently, in vivo experimental studies have revealed that those misfolded proteins can behave in a prion-like manner, as the inoculation of misfolded α-syn or tau aggregates induce the native endogenous proteins to adopt an abnormal conformation in the mouse brains[11–13]. In those inoculation experiments, it takes several months after inoculation for the mice to show endogenous abnormal protein accumulations, which are usually phosphorylated. The pathological studies of those experiments suggest that neuronal connections are involved in the spread of seeds, yet this has not been demonstrated directly in vivo.

We previously studied the effect of callosotomy (dissection of corpus callosum) on α-syn pathology after the intrastriatal inoculation of α-syn preformed fibril (PFF) seeds, revealing the involvement of corpus callosum in seed spreading to the contralateral hemisphere. Unexpectedly, this spreading of seeds to the contralateral hemisphere occurred within 24 h after the inoculation of the seeds[14]. It was recently reported that neuronal activity modulates α-syn aggregation and spreading in hippocampus[15]. Based on these results, in the current study, we labeled PFF seeds with quantum dots to visualize them directly and developed an acute slice system to observe the movement of seeds along the corpus callosum. Using this system, we could observe the pharmacological effects on the acute phase of seed spreading and found this spreading is dependent on neuronal activity[14,16–19].

## Results

### Quantum dot-labeled α-syn seeds for molecular imaging.
To visualize α-syn seeds, we used CdSeTe quantum dot (QD) labeling because it is very bright and shows less photobleaching than other labels. Its emission maximum is ~705 nm, and since fluorescence emission at 700 nm shows little autofluorescence, this condition is suitable for tracking small QD-labeled seeds in thick tissue of the brain slices. To label α-syn seeds with QDs, we coupled α-syn-PFFs and QDs-COOH using the amine coupling reaction (Supplemental Fig. S1a). The labeled PFFs were confirmed with electron micrograph (EM) analysis (Supplemental Fig. S1b) and ThT analysis (Supplemental Fig. S1c). After the sonication of QDs-labeled α-syn-PFFs (QD-α-syn-PFFs), they were used for seeds (QD-α-syn seeds) and injected to the mouse

striatum of the right hemisphere as described previously[14]. When frozen coronal sections were obtained from seed-injected mouse brain and immunostained with anti-α-syn antibody, all QD-α-syn-seeds in all regions of interest (ROI) at 1 h after injection were positive for α-syn immunoreactivity (Supplemental Fig. S1d and e), confirming that QD fluorescence shows α-syn seeds.

Cytotoxicity of QD-α-syn seeds was tested using neuronal cells (Supplemental Fig. S2). When we administered α-syn seeds or QD-α-syn seeds to neuro2a cells and cell viabilities were examined by MTT assay, there was no statistical difference in their viability between non-labeled α-syn and QD-α-syn seeds (Supplemental Fig. S2), suggesting no effect of QD-labeling on cellular viability.

### α-syn seeds are associated with intracellular trafficking-related molecules.
We then immunohistochemically examined the association of QD-α-syn seeds with molecular markers for the organelles and intracellular trafficking using frozen sections (Fig. 1a). Slices from the mice injected with QD-labeled α-syn monomer (QD-sa-α-syn monomer) were used as controls. The early endosome marker EEA1[20] colocalized with QD-α-syn seeds after injection (Fig. 1b, Supplemental Fig. S3a). LAMP1 and LAMP2 were used as lysosomal markers[20] and colocalized with QD-α-syn seeds in all ROI, though less in the corpus callosum for LAMP1 (Fig. 1c, d, Supplemental Fig. S3b, c). SNARE-protein synaptobrevin-2/vesicle-associated membrane protein 2 (VAMP2) was used as α-synaptic vesicular marker[21] and colocalized with QD-α-syn seeds mainly at the corpus callosum (Fig. 1e, Supplemental Fig. S3d). Amyloid precursor protein (APP), a protein related to one of the AD genes and an axonal transport marker[22], colocalized with QD-α-syn seeds similar to VAMP2 (Fig.1f, Supplemental Fig. S3e). Leucine-rich repeat kinase 2 (LRRK2), a protein related to one of the PD genes[23], colocalized with QD-α-syn seeds similar to other markers (Fig.1g, Supplemental Fig. S3f). We could observe the signals of QD-labeled α-syn monomer only in the ROI including the injected area (Fig. S4a-f) and these signals show no colocalization with any markers, suggesting QD-labeled monomers did not spread to the other ROI and could not be incorporated into the cell (Supplemental Fig. S4a–f). These results suggest QD-α-syn seeds colocalized and moved with intracellular trafficking-related molecules.

We also investigated whether QD-α-syn seeds could be detected in regions of the contralateral hemisphere, where cell-to-cell transmission is necessary for seeds to be transferred. QD-α-syn seeds appeared in medium spiny neurons in the contralateral striatum, and in neurons in the substantia nigra pars compacta (SNc), entorhinal cortex (EC), and amygdala (Amy) at 6 h after the injection of seeds (Fig. 1j, k, n, o), although QD-sa-α-syn monomers were not found in these regions (Fig. 1h, i, l, m). The results suggest QD-α-syn seeds were transmitted from cell to cell through neuronal networks.

### Visualization of QD-α-syn seed dynamics in acute brain slices.
Because our previous study suggested that α-syn seeds rapidly disseminate to the contralateral hemisphere through the corpus callosum[14], we decided to visualize the trafficking of QD-α-syn seeds using acute slices. We prepared living acute brain slices 1 h after injection of QD-α-syn seeds (Fig. 2a). With these slices, we could observe and record the movement of QD-α-syn seeds. After the recording of the QD-α-syn seed movement around the corpus callosum of the contralateral hemisphere (Supplemental Movie 1), tracking analysis for multiple molecular dynamics was performed (Fig. 2b and c). The speed of QD-α-syn seeds was faster in corpus callosum than in cortex and striatum (Fig. 2d)

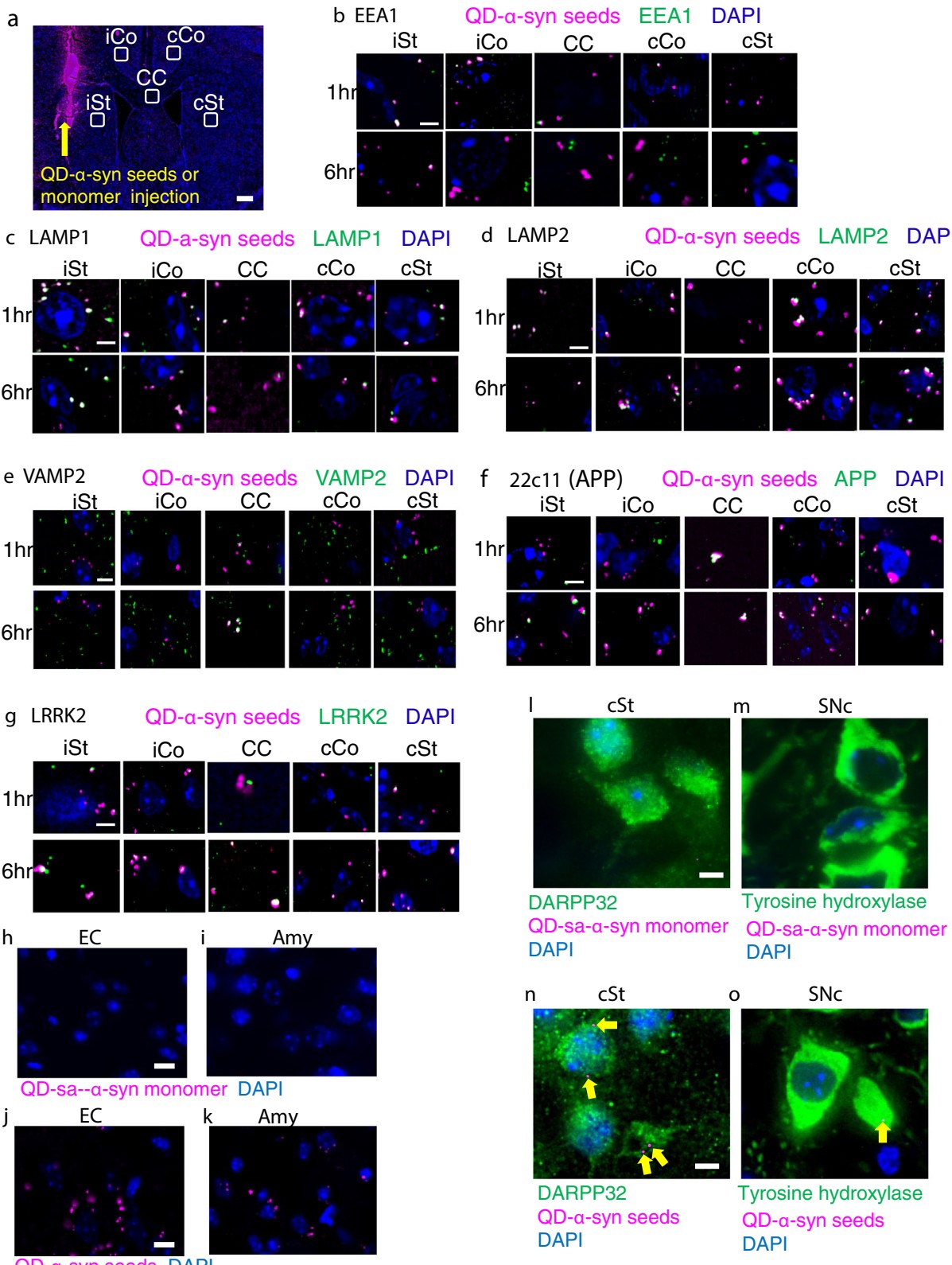

and the mean speed was $0.28 \pm 0.04\,\mu m/s$ (Supplemental Fig. S10c). We found that QD-α-syn seeds migrated into the contralateral cortex and striatum (Fig. 2e, right panel), whereas no signal was found when QD only (i.e. without α-syn seeds) was injected (Fig. 2f, right panel). We also tested another labeling method using Qdot 705 streptavidin(sa) conjugate to examine whether there was a difference between QD-sa-α-syn monomer

(Supplemental Fig. S5a and b) and QD-sa-α-syn seed migration in the living slice. Tracking analyses for QD-sa-α-syn monomer in the slice showed no migration signals in the contralateral ROI (Supplemental Fig. S5c, upper right panel), whereas α-syn seeds which were labeled with QD by the same method (QD-sa-α-syn seeds) migrated in the contralateral region. (Supplemental Fig. S5c, lower right panel).

**Fig. 1 Characterization of QD-α-syn seeds. a** Regions of interest (ROI) (white rectangles) for immunohistochemical stainings with antibodies at 1 and 6 h after QD-α-syn-seed or QD-sa-α-syn-monomer injection. iSt ipsilateral striatum (injection side), iCo ipsilateral cortex, CC corpus callosum, cCo contralateral cortex (opposite side of injection), cSt contralateral striatum. Bar: 500 μm (**a**). **b** QD-α-syn-seeds (purple) and anti-EEA1 immunoreactivity (green). White: colocalization. Blue: DAPI. **c**–**g** Other markers (green): LAMP1 (**c**), LAMP2 (**d**), VAMP2 (**e**), 22C11 for APP (**f**), LRRK2 (**g**). Quantitative analyses are shown in Supplemental Fig. S3. **h** No QD-sa-α-syn monomer in entorhinal cortex(EC). **j** QD-α-syn seeds in EC. **i** No QD-sa-α-syn monomer in Amygdala (Amy). **k** QD-α-syn seeds in Amy. **l** No QD-sa-α-syn monomer in DARPP32 positive medium spiny neurons in cSt. **n** Localization of QD-α-syn seeds in DARPP32 positive medium spiny neurons in cSt. **m** No QD-sa-α-syn monomer in tyrosine hydroxylase positive neuron in SNc. **o** Localization of QD-α-syn seeds in tyrosine hydroxylase positive neuron in SNc. Yellow arrows: QD-α-syn seeds in cSt (**n**) and SNc neurons (**o**). Bar: 5 μm (**b**–**o**).

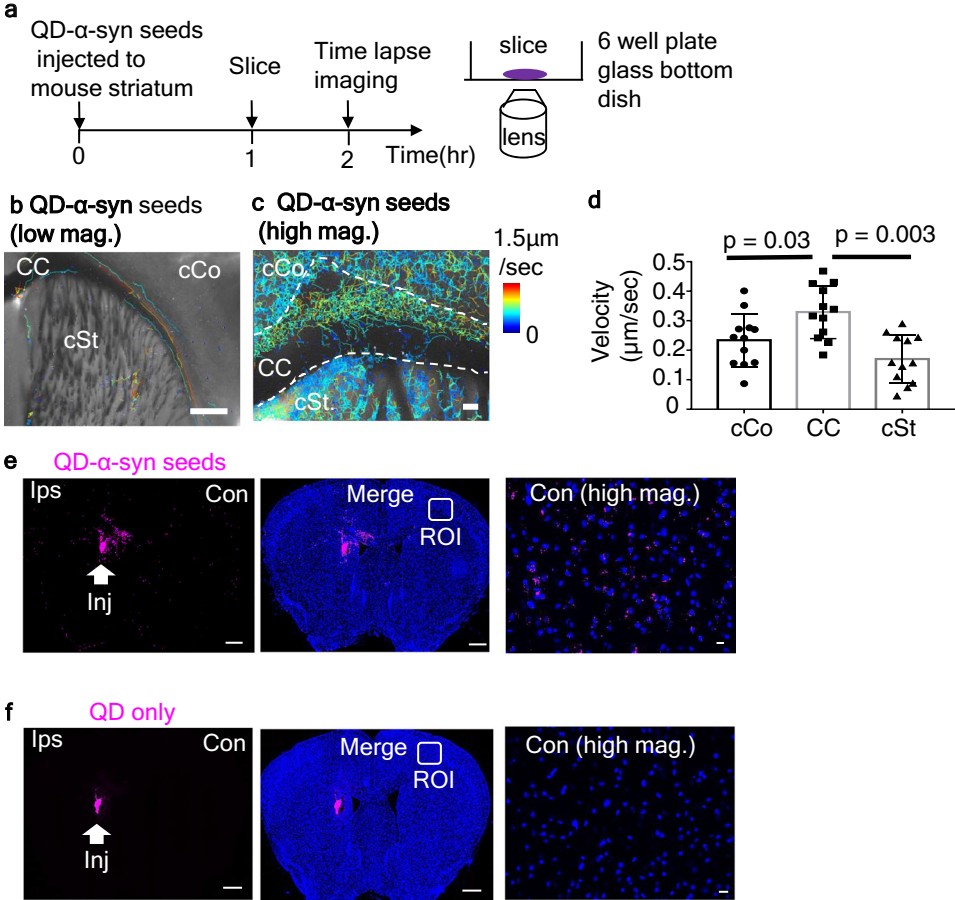

**Fig. 2 QD-α-syn seed dynamics and localization in the brain slices. a** Experimental procedures for time lapse imaging of slice. Slices were half submerged in culture medium and tightly attached at the bottom of six-well plate glass bottom dish. Recording was performed in humidified cultured box saturated with 95% $O_2$/5% $CO_2$. **b** Tracking of α-syn seeds (white dots) conducted by Image-J plug-in Trackmate. Velocity shown by line color. Bar: 500 μm. **c** High magnification of α-syn seed tracking. Color bar shows speeds. Bar: 50 μm (see Supplemental video 1 for their dynamics). **d** Velocity of QD-α-syn seeds in each area of (**c**). **e** Fluorescence view of tissue sections (10 μm thick) at 2 h after injection of QD-α-syn seed. **f** QD only was injected. Indicated squares in the contralateral hemisphere (middle panels) were magnified (right panels) (**e**, **f**). Bar: 500 μm (low mag.: left and middle panels), 10 μm (high mag.: right panel).

These findings suggest that α-syn seeds are transported from the ipsilateral hemisphere of the injected side to the contralateral cortex and striatum through corpus callosum by axonal transport.

**Pharmacological effect on α-syn seed dynamics**. To characterize trafficking dynamics of α-syn seeds, we examined the pharmacological effects on QD-α-syn seed numbers counted in the ROI and their trafficking speed (Fig. 3a). Because the ROI was selected at the exit to the contralateral hemisphere of the corpus callosum, we suppose that the number observed reflected the efficiency of the seed spreading to the contralateral hemisphere. Results summary was also shown in Table 1. First, we examined the effect of dynasore, an inhibitor of clathrin and dynamin-mediated

endocytosis. The dynasore significantly reduced QD-α-syn seed number (Pre: 100%, Dyn: 25.9 ± 3.6%, reduction rate = seed number reduction after dynasore treatment compared with that of pretreatment (Pre): 74.1 ± 3.58%, $p = 0.0001$; recovery rate = increased ratio of seed number after washout compared with that of dynasore treatment: 58.5 ± 15.5%, $p = 0.042$, $n = 5$ mice) (Fig. 3b), suggesting that α-syn trafficking was increased by the endocytosis of α-syn seeds into neurons. Second, we tested the effect of colchicine, which inhibits microtubule polymerization and fast axonal transport[24]. The colchicine significantly reduced QD-α-syn seed number (Pre: 100%, Col: 31.6 ± 4.1%, reduction rate: 68.4%, $n = 7$ mice) and slowed its speed (Fig. 3c). The result suggests that the trafficking of α-syn seeds is dependent on fast axonal transport in association with microtubules.

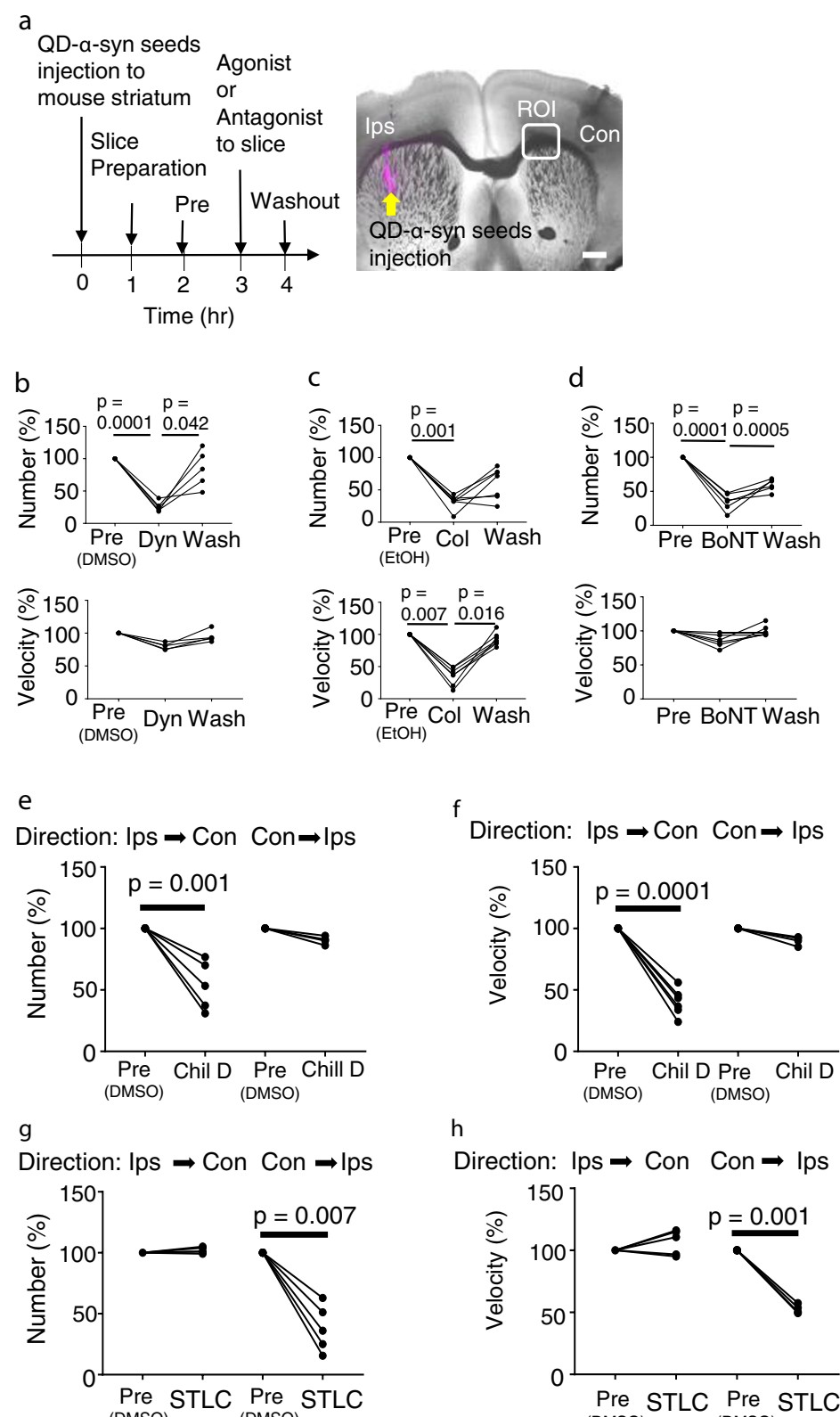

Third, we examined the effect of botulinum toxin (BoNT), which prevented neurons from releasing synaptic vesicles of neurotransmitters by cleaving SNARE proteins[25,14]. The BoNT significantly reduced rates of QD-α-syn seed number (Pre: 100%, BoNT: 34.7 ± 5.1%, reduction rate: 65.3%, $n = 7$ mice Fig. 3d). These results suggest injected α-syn seeds were incorporated into the neurons by clathrin-dependent endocytosis and transported on the fast axonal flow and may be affected by the inhibition of neurotransmitter release.

**Retrograde and anterograde axonal transport is involved in seed spreading.** Dynein and kinesin are key molecules for retrograde and anterograde axonal transport, respectively. To examine which axonal transport is involved in α-syn seed

**Fig. 3 Pharmacological analysis of QD-α-syn seed dynamics. a** Experimental procedure for pharmacological analysis on acute QD-α-syn seed dynamics. Ips, ipsilateral. Con contralateral. Yellow arrow: QD-α-syn seeds injected to striatum. ROI: Region of interest. Bar: 500 μm. **b–d** Effect of drugs: **b** dynasore (Dyn); **c** colchicine (Col); **d** botulinum toxin (BoNT). Top: QD-α-syn seed number normalized by the number at pretreatment (Pre) with solvent only. For dynasore, DMSO was used for dilution of dynasore (**b**), and EtOH was used for dilution of colchicine (**c**). The same concentration of DMSO or EtOH were added in the solution for pretreatment. Bottom: mean velocity of QD-α-syn seeds normalized by that of pretreatment. Each line indicates the result from the same slice, respectively (n = 4–6 mice). **e–h** Drug effects on the axonal transport of QD-α-syn-seeds. **e, f** Effect of Chil D, dynein inhibitor (retrograde transport inhibitor) on QD-α-syn seeds transport (**e** α-syn number, **f** mean velocity). **g, h** Effect of STLC, kinesin inhibitor (anterograde transport inhibitor) on QD-α-syn seeds transport (**g** α-syn number, **h** mean velocity). n = 4–6 mice (Chil D), n = 5 mice (STLC). p < 0.05: statistically significant by one-way ANOVA followed by Tukey's post test.

**Table 1 Results summary for pharmacological effects on dynamics of α-syn seeds.**

| Drugs | Results summary | | | |
|---|---|---|---|---|
| | Effect | Slices (in vitro) | | In vivo |
| | | Number of seeds | Velocity | |
| Dynasore | Inhibitor of clathrin-dependent endocytosis | ↓↓ | → | |
| Colchicine | Inhibitor of Microtuble polymerization | ↓↓ | ↓↓ | |
| BoNT | Blockade of neurotransmitter release | ↓↓ | → | |
| Chill D | Inhibitor of retrograde transport | ↓↓ | ↓↓ | |
| STLC | Inhibitor of anterograde transport | ↓↓ | ↓↓ | |
| GABA | GABA receptor agonist | ↓ | → | |
| Bicuculline | GABA$_A$ receptor antagonist | ↑ | → | |
| Ampakine | Positive allosteric modulator | ↑↑ | → | |
| DNQX | AMPA receptor antagonist | ↓↓ | → | |
| AP5 | NMDA receptor antagonist | ↓↓ | → | |
| TTX | Inhibitor of voltage-gated Na channel | ↓↓ | → | |
| Riluzole | Inhibitor of NMDAR and Kainate-R and Na channel | ↓↓ | → | ↓↓ |
| Perampanel | AMPA receptor antagonist | ↓↓ | → | ↓ |
| Sertraline | Inhibitor of clathrin-dependent endocytosis | ↓↓ | → | ↓ |
| Rifampicin | Inhibitor of Aβ oligomerization | → | → | ↓ |

↓or ↑: p < 0.05; ↓↓ or ↑↑: p < 0.01 against pre or control; →: no significant difference.

spreading, we used dynein and kinesin inhibitors (Fig.3e–h). Dynein inhibitor (Chil D) significantly and exclusively reduced the number and speed of α-syn seeds moving from the ipsilateral to contralateral hemisphere (Number: from Ips to Con, Pre, 100%,Chil D, 53.6 ± 8.91%, from Con to Ips, Pre, 100%, Chil D, 90.3 ± 1.61%, Velocity: from Ips to Con, Pre, 100%,Chil D, 40.0 ± 4.50%, from Con to Ips, Pre, 100%, Chil D, 89.9 ± 3.53%, Fig. 3e and f). Conversely, kinesin inhibitor (STLC) significantly and exclusively reduced the number and speed of α-syn seeds moving in the contralateral to ipsilateral direction (from Con to Ips, number: Pre, 100%, STLC, 38.1 ± 8.58%, velocity: Pre, 100%, STLC, 52.7 ± 1.82%, Fig. 3g and h). These findings suggest that in this ROI the seed movement from ipsilateral to contralateral hemisphere is driven by retrograde transport and the reverse direction is driven by anterograde transport.

**Spreading of α-syn seeds depends on neuronal activity.** Cortical neurons projecting to striatal neurons are glutamatergic[26]. To examine whether α-syn seed dynamics are dependent on neuronal activity, we modulated synaptic transmission and observed α-syn seed dynamics. Each agonist or antagonist for glutamatergic or GABAergic synaptic transmission was bath-applied. GABA agonist (GABA, Fig. 4a), AMPAR antagonist (DNQX, Fig. 4d), and NMDAR antagonist (AP5, Fig. 4e) significantly reduced the migration number of α-syn seeds. Conversely, GABA antagonist (bicuculline, Fig. 4b) and a positive modulator of AMPAR (Ampakine, Fig. 4c) increased the migration number of α-syn seeds without significant difference in mean speed. These findings suggest that the migration number of α-syn seeds was regulated by neuronal activity. We further confirmed this by blocking

neuronal activity with tetrodotoxin (TTX), which is known to block sodium channels; this reduced the migration number of α-syn seeds (Fig.4f).

**The effect of clinically available drugs on α-syn seed dynamics.** Neuronal activities are regulated by some clinically available drugs. We tested the effect of the following clinically used drugs on α-syn seed migration: (1) riluzole, which is used for the treatment of ALS, and inhibits NMDA-R and kainate-R[27] as well as voltage-gated sodium channels in damaged neurons[28], (2) perampanel, which is used for the treatment of epilepsy and is an AMPAR antagonist[29], (3) sertraline, an antidepressant known as a selective serotonin reuptake inhibitor (SSRI) and reported as an inhibitor of clathrin-dependent endocytosis[30], and (4) rifampicin, which is an antibiotic used to treat several types of bacterial infections, including tuberculosis, and reported as an inhibitor of oligomer formation and fibrilization of amyloids[31]. Riluzole, perampanel, and sertraline significantly reduced the number of migrating α-syn seeds without changing migration speed (Fig. 5a–c). However, rifampicin did not significantly change the number and speed of α-syn seeds (Fig. 5d). These findings suggest that certain drugs could reduce the dissemination of α-syn seeds, resulting in decreased α-syn pathology.

**The reduction of α-syn pathology by clinically available drugs.** Because clinically available drugs could reduce migrating α-syn seed numbers, we examined the effect of those drugs on in vivo transmission. As we previously reported, a pathological study was performed 1.5–2 months after the injection of α-syn seeds in the striatum. Each of the four drugs mentioned above was

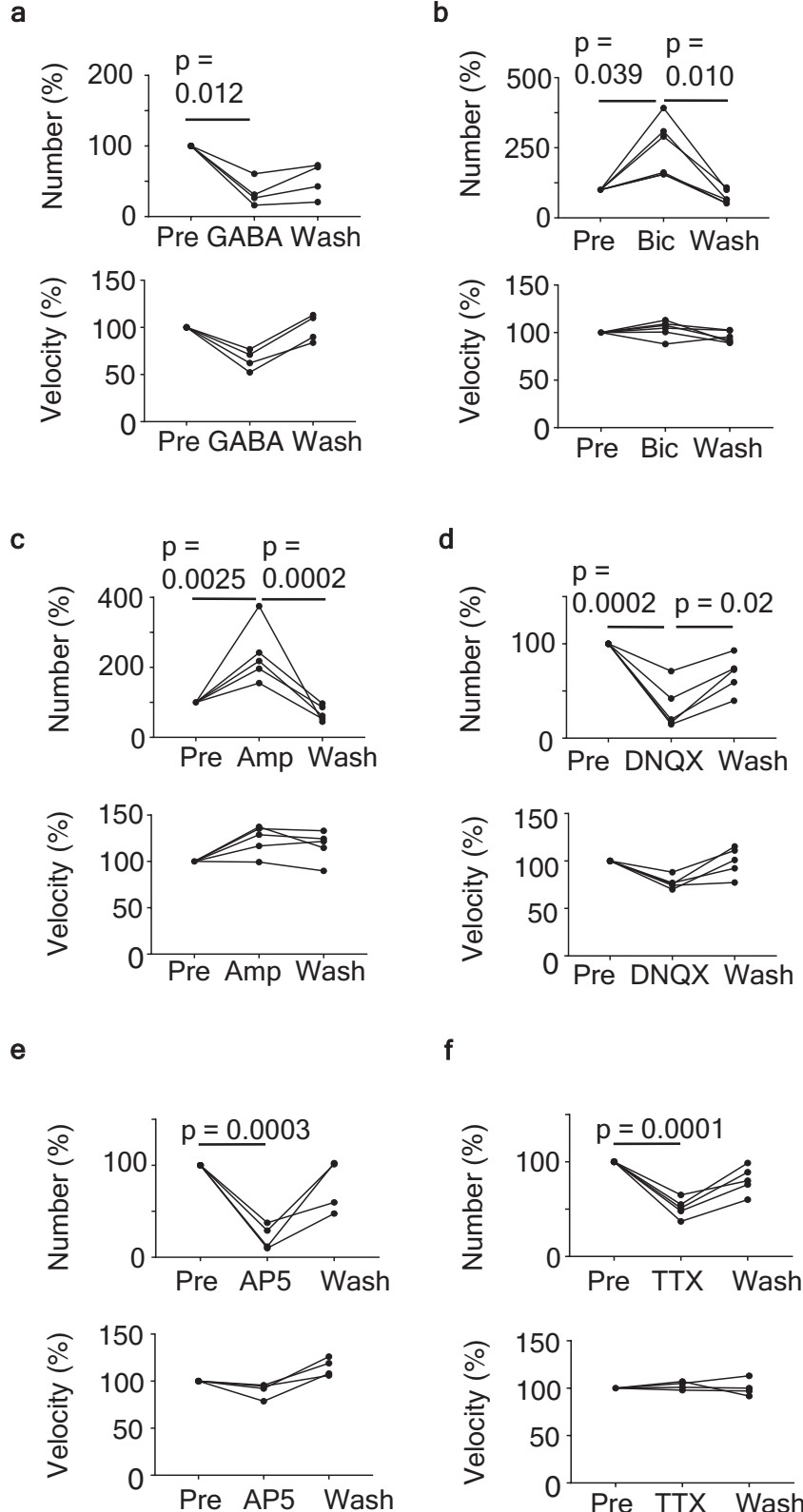

**Fig. 4 QD-α-syn seed dynamics depend on neuronal activity.** Effect of drugs bath-applied on QD-α-syn seed dynamics in the slice. **a** GABA, **b** bicuculline (GABAa receptor antagonist), **c** ampakine (AMPA receptor agonist), **d** DNQX(AMPA receptor antagonist), **e** AP5 (NMDA receptor antagonist), and **f** TTX, tetrodotoxin. Top panels: number of QD-α-syn seeds normalized by pretreatment. Bottom panels: mean velocity of QD-α-syn seeds normalized by that of pretreatment. Experimental procedure and ROI are the same as in Fig. 3a. $n = 4$–6 mice. $p < 0.05$: statistically significant by one-way ANOVA followed by Tukey's post test.

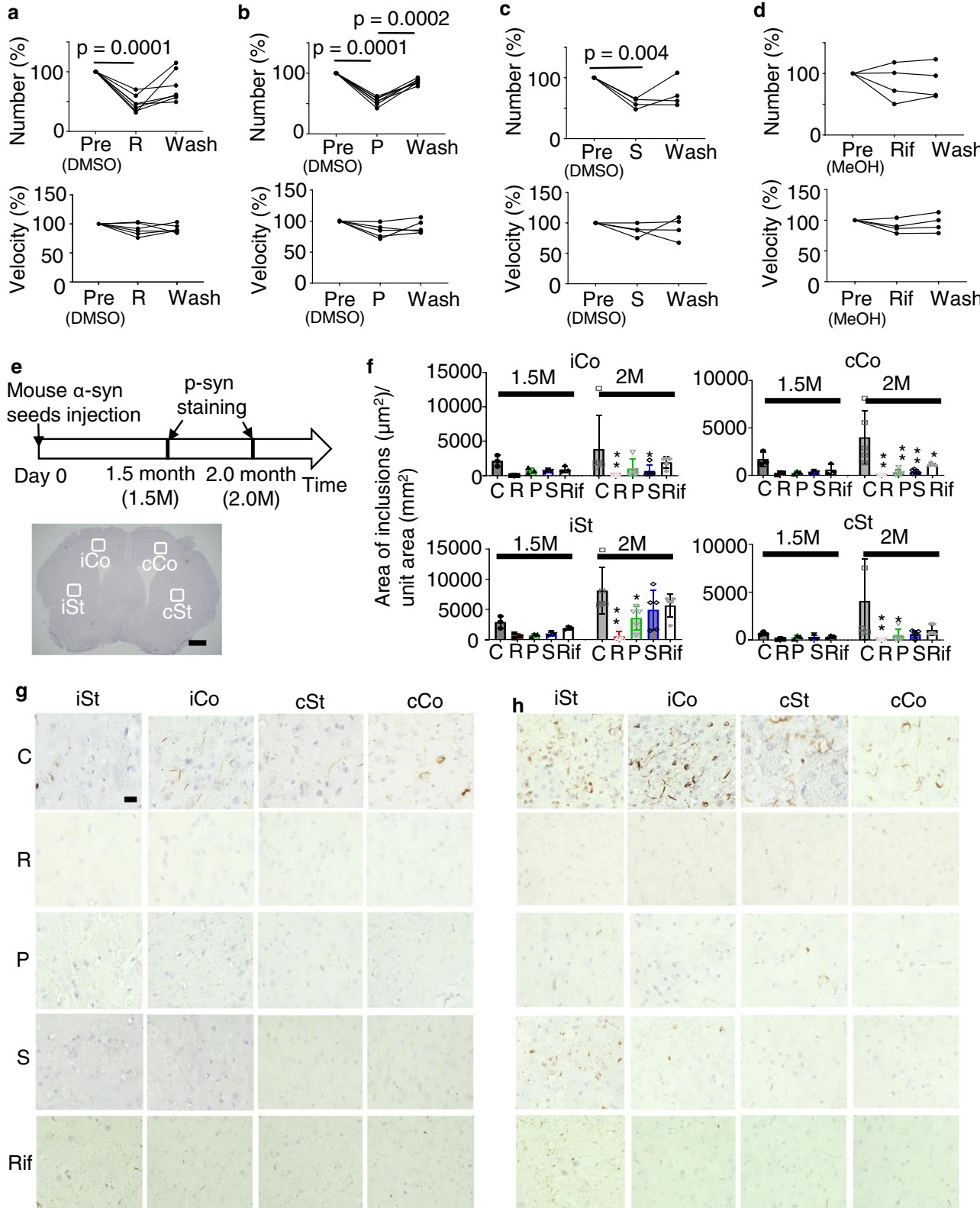

**Fig. 5 Effect of clinically available drugs on α-syn seed dynamics in the acute slice and transmission experiment. a–h** The effects on α-syn seed numbers (top) and mean velocity(bottom) in acute slice assay by riluzole(R) (**a**), perampanel(P) (**b**), sertraline(S) (**c**), and rifampicine(Rif) (**d**). **e** Experimental procedure for chronic drug effects on phosphorylated α-syn (p-syn) pathology after mouse α-syn seeds injection (in vivo transmission experiment). ROI iSt ipsilateral striatum, iCo ipsilateral cortex, cCo contralateral cortex, cSt contralateral striatum. Bar: 500 μm. **f** Quantitative analyses for p-syn expression in mice ($n = 3$ mice, respectively). *$p < 0.05$, **$p < 0.01$ compared to C. C control (injection of PBS), Pathology was examined at 1.5 M **g** and 2.0 M **h** after mouse α-syn seed injection. Bar: 10 μm.

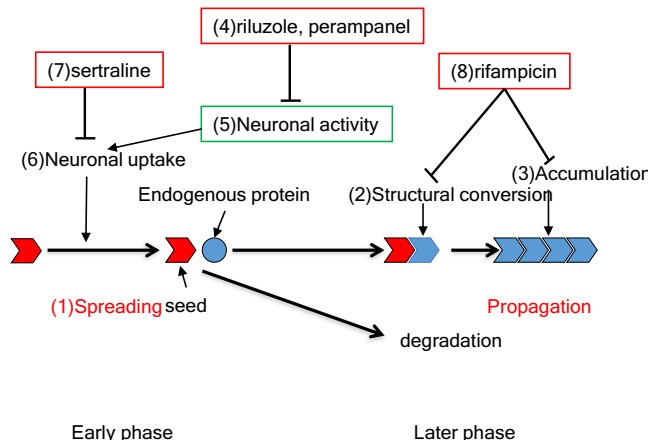

**Fig. 6 Schematic view of drug effects on α-syn seed transmission.** In the transmission experiment used in this study, seeds are disseminated rapidly (1). Those seeds convert endogenous protein to a misfolded version (2), which could be visible several months later (3). Some drugs such as riluzole and perampanel (4) reduce neuronal activity (5) and attenuate the uptake of seeds (6). Sertraline (7) may reduce seed uptake directly. Rifampicin (8) does not affect this early phase, but reduces structural conversion (2) or accumulation (3) at a later phase.

administered from 7 days before the inoculation of seeds until the animal was killed, and synuclein pathology was detected by antibody to phosphorylated α-syn (p-syn) (Fig. 5e). Compared to control mice, those that received riluzole showed remarkably reduced p-syn immunoreactivity at every ROI (Fig. 5f–h and Supplemental Fig. S6). Treatment with riluzole after α-syn seed inoculation showed no effect (Supplemental Fig. S7a–d), suggesting that α-syn seed dissemination occurred in a short time after the inoculation and this early α-syn seed uptake phase is regulatable by riluzole, but once seeds have disseminated, riluzole cannot inhibit the propagation (Fig. 6). Other drugs including rifampicin also reduced p-syn pathology, but less effectively (Fig. 5f–h).

To further confirm the prevention of p-syn pathology by riluzole biochemically, we examined sarkosyl insoluble fractions by western blot and filter trap assay (FTA), which is used for the detection of abnormal protein aggregation (Supplemental Fig. S8c). Western blot and FTA revealed that sarkosyl insoluble fractions from mouse brains at 2 months after α-syn seed injection showed pathological p-syn aggregates in the brains of control or those treated with riluzole after injection (i.e., +Ril (a)), whereas almost no aggregation was observed in those treated with riluzole before α-syn-seed injection (i.e., +Ril (b)) (Supplemental Fig. S8a, b, d).

Finally, we examined whether QD-labeled α-syn-seeds hold the seeding ability by investigating p-syn pathology at 2 months after QD-α-syn-seed injection. The QD-α-syn seeds clearly showed positive p-syn immunoreactivity in both Ips and Contra regions in the brain (Supplemental Fig. S9). Altogether, these findings suggest that α-syn-seeds (with or without QD-labeling) have the ability to induce p-syn pathology, and riluzole pre-treatment before α-syn-seed inoculation is most effective in alleviating this synuclein pathology.

**Amyloids, in general, may spread in the same manner as α-syn seeds.** Recently, the prion-like spreading of misfolded protein accumulations was suggested not only for synuclein but also for tau and Aβ[12,13,32,33]. Therefore, we examined whether $Aβ_{1-42}$-PFF and tau-PFF spread like α-syn seeds did in the acute slices. QD-labeled $Aβ_{1-42}$-PFF and tau-PFF moved in the acute slices as

did α-syn seeds (Supplemental Fig. S10a and b). Moving speeds of $Aβ_{1-42}$-PFF and tau-PFF were similar to α-syn seeds (Supplemental Fig. S10c). Furthermore, riluzole significantly reduced the migrating number of $Aβ_{1-42}$-PFF and tau-PFF (Supplemental Fig. S10d and e). To confirm that the spreading was not due to the QD-labeling, we examined the spreading of Alexa 488-labeled $Aβ_{1-42}$-PFF and tau-PFF (Supplemental Fig. S11a and b). Alexa488-labeled $Aβ_{1-42}$-PFF or tau-PFF also spread to contralateral cortex and striatum, similar to QD-labeled ones. These findings suggest a general mechanism among amyloid proteins for PFF spreading.

**Comparison of dynamics for non-pathogenic physiological proteins and α-syn seeds.** Finally, we tested whether non-pathogenic physiological proteins showed prion-like transmission. Tubulin and microtubules were used as non-pathogenic physiological proteins in this experiment. Microtubule formation by tubulin polymerization was monitored at $OD_{340}$ and confirmed (Supplemental Fig. S12a). Microtubules were fixed with glutaraldehyde to prevent depolymerization. Result of western blot showed that fixed microtubules were detected as larger size proteins (>245 kDa) than tubulin monomers (Supplemental Fig. S12b). Then, the tubulin monomers and microtubules were labeled with QD as QD-α-syn seeds and injected into mice striatum (Supplemental Fig. S12c, upper left panel). Compared to the movement of QD-α-syn seeds in the ROI, QD-tubulin showed no signal and QD-microtubules showed very few signals in the ROI (Supplemental Fig. S12c and d). These finding suggest that physiological proteins such as tubulin monomers and fibrils such as microtubules were not incorporated into the cell as α-syn-seeds.

## Discussion

Accumulating evidence suggests that, in neurodegenerative disorders such as synucleinopathy and tauopathy, disease-related proteins exhibit properties of template-driven self-assembly and their aggregates transmit between cells and convert cognate monomers into an ordered assembly[6,34–36]. This prion-like transmission mechanism is supported by the fact that in AD and PD, the tau and synuclein pathology expands from one region to connected regions[7,9]. Experimentally, when pathological tau or synuclein protein aggregates derived from diseased brains or formed from synthetic protein are injected into rodent brains, they induce accumulations of the corresponding endogenous proteins[37,38]. In those experiments, the pathology seems to expand through the brain connectome[39,40]. However, direct observation of injected protein seed spreading was not performed in vivo.

Our previous study and others suggested that very rapid dissemination of seeds occurs in vivo within 24 h after the seed injection[14,41] and we thought the spreading process might be observable with labeled seeds. In this study, we labeled α-syn PFFs with QD, which are inorganic fluorescent nanocrystals that provide a useful alternative for studies that require long-term and multicolor imaging of cellular and molecular interactions[42]. We successfully observed α-syn seeds spreading in acute living slices and found regulatory factors for it.

After injection into the striatum, we know that α-syn seeds move to the contralateral hemisphere through the corpus callosum because callosotomy significantly impeded this process, as reported previously[14]. We therefore set an ROI at the exit of the corpus callosum to the contralateral hemisphere. Due to the strong QD fluorescence, we could use thick slices, which keep the neural network structure including corpus callosum. Then, we found that α-syn seeds move on the corpus callosum, which is a

bundle of axons, and the mean speed of α-syn seed movement was 0.28 ± 0.04 μm/s, which is compatible with fast axonal transport. Moreover, colchicine, an inhibitor of fast axonal flow, inhibited the migration of α-syn seeds, supporting the idea that α-syn seeds move on fast axonal transport. A previous study using mouse neurons cultured in microfluidic devices showed that α-syn fibrils were transported by anterograde axonal transport[43]. Our pharmacological study revealed that α-syn seed movement from the ipsilateral to the contralateral hemisphere was on the dynein-dependent, retrograde transport and its movement in the reverse direction is on kinesin-dependent, anterograde transport. Dynein and kinesin transport are fast axonal transport and the mean speed is 0.3–0.7 μm/s[44,45], compatible with our result. Crossed-corticostriatal neurons were reported[46] and those neurons, though not all, might be involved in those transports. In our study, APP and VAMP2 colocalized with QD-α-syn seeds mainly in the corpus callosum. Previous reports indicated the APP was required for kinesin-mediated axonal transport[47]. However, vesicles containing APP move bi-directionally and require dynein and kinesin[48]. Although VAMP2 is a member of the vesicle-associated membrane protein (VAMP)/synaptobrevin family and is thought to participate in neurotransmitter release at the synapse, VAMP2-containing carrier vesicles are transported by the anterograde axonal transport motor KIF1A[49]. Thus, the immunohistochemical study also reflects the α-syn seeds on axonal transport machinery.

Because we established a monitoring system for α-syn seeds spreading at the exit of the corpus callosum as an ROI, the migration number of α-syn seeds in the ROI is regulated by entrance of seeds. The entrance of seeds into the ROI could be regulated by the number of seeds taken into neurons. Dynasore, an inhibitor of clathrin and dynamin-mediated endocytosis, reduced the number of α-syn seeds, suggesting the entrance of seeds into neurons could be mediated by clathrin-dependent endocytosis, which is a dominant mode of synaptic vesicle retrieval at physiological stimuli[50]. There is another mode, activity-dependent bulk endocytosis, which is triggered during elevated neuronal activity[51]. In our pharmacological study, an agonist or antagonist for glutamatergic or GABAergic synaptic transmission was applied; the excitatory effect increased the number of migrating α-syn seeds while the inhibitory effect decreased it, revealing that the entrance of seeds into neurons is regulated by neuronal activities and reduced neuronal activity could decrease the entrance of seeds.

Based on those results, we tried to find clinically available drugs that can decrease the spread of α-syn seeds using our system. Riluzole, an inhibitor of NMDA-R and kainate-R[27] reduced the number of α-syn seeds most effectively. Perampanel, an AMPAR antagonist[29], and the SSRI sertraline, an inhibitor of clathrin-dependent endocytosis[30], also reduced the numbers of migrating α-syn seeds in the ROI, but not as effectively. If the dissemination of α-syn seeds is regulated by the endocytosis of the seeds, and that process is dependent on neuronal activity, then activity-dampening drugs should decrease α-syn pathology with chronic administration. As expected, riluzole most effectively suppressed the α-syn pathology detected with an antibody to phosphorylated α-syn. In a previous study, we hypothesized two different modes of α-syn seed transmission and propagation. In the step-by-step mode, misfolded α-syn propagation forms aggregates, and then seeds are released in a neuron, and finally, the seeds are transmitted to a connected neuron via the synapse. In another mode called dissemination mode, seeds are directly disseminated transneuronally and induce the propagation of misfolded α-syn to form aggregates in the disseminated area[14]. In the most prion-like transmission experiments, seeds are injected into a certain region, and the α-syn pathology detected as p-syn deposition was

evaluated a few months to several months later. Callosotomy before injection of α-syn seeds reduced the α-syn pathology but not callosotomy 1 day after injection, suggesting the dissemination must have occurred rapidly in our previous experiment[14]. The present study further confirmed that rapid spreading through corpus callosum occurred and reducing this spreading by decreased intake of α-syn seeds with suppression of neuronal activity could alleviate the α-syn pathology. Rifampicin was reported as an inhibitor of oligomer formation and fibrilization of amyloids and no effect was observed in the number of migrating α-syn seeds in our slice experiment. However, the α-syn pathology was reduced by rifampicin to a similar level as sertraline. The result suggests rifampicin works on the propagation of α-syn (amyloid formation), but not on the spreading. Our experiment could dissect the prion-like transmission process into spreading and propagation; only the latter was suppressed by rifampicin (Fig.6). It is noteworthy that the latter phase could be controlled by the degradation of seeds and conversion of endogenous proteins into the misfolded ones.

Previous studies indicate that riluzole improves MPTP-induced movement disorder in a rat model of multiple system atrophy (MSA)[52] and a marmoset model of PD[53]. However, clinical trials of riluzole administration to symptomatic PD and MSA patients showed no significant improvement[54,55]. In those trials, symptomatic evaluations were performed after short-term administration such as 1 or 4 weeks. Riluzole is an FDA-approved drug for ALS and confirmed to prolong the survival of ALS patients. This effect is thought to be due to the anti-excitotoxic effect of riluzole. However, considering a common feature of misfolding protein aggregation in neurodegenerative disorders, TDP-43 aggregation spreading may be inhibited by riluzole as discussed later for Aβ and tau amyloid. Early-phase and long-term trials are necessary for the evaluation of riluzole on synucleinopathy. Furthermore, the selection of a specific regulator for the activity of the neural networks susceptible to a specific disease should be important for a successful trial. On the other hand, many drugs could have the unexpected effect on the neuronal activities. Drugs discovered as effective in the transmission experiment should be examined whether those are effective on the early phase of transmission or the later phase.

Although we did not use the extracts from the disease brain as seeds for the assay system because the quality of the diseased tissue varies widely across cases, it is important to confirm whether these extracts show spreading based on the same mechanism. We believe that for the drugs discovered by our assay system, it is better to observe their effects on transmission with extracted pathogenic seeds from disease brain to create a foundation for future translational work.

Our results show that α-syn seed trafficking uses a rather non-specific endocytic pathway. Thus, we wondered if other amyloids also spread through this machinery. As expected, QD-Aβ PFFs and QD-tau PFFs are migrating in the ROI as α-syn seeds at the same migration speed. On the other hand, physiological fibrils such as microtubules were much less incorporated than α-syn seeds, suggesting some amyloid-specific mechanism of cellular uptake might exist. Riluzole also decreased the number of Aβ PFFs and tau PFFs in the ROI. Proof-of-concept experiments for other amyloidogenic disease proteins using our system should be possible to screen for effective drugs for those diseases.

## Methods

**Preparation of preformed fibrils**. Preformed fibrils (PFFs) in this study were prepared as follows.

1. *Human and mouse α-syn PFFs*: Human and mouse α-syn seeds were prepared as described previously[14]. In brief, the *Escherichia coli* strain BL21(DE3) was transformed with the expression vector pET15b encoding

human or mouse α-syn. The expression of His-tagged α-syn was induced by the addition of 0.5 mM isopropyl β-D-thiogalactoside at 37 °C for 3 h. Cells were lysed by ultrasonication in PBS containing 2% Triton X-100, centrifuged at $20,000 \times g$ for 30 min, and the supernatant was then loaded on a Ni Sepharose 6 Fast Flow column (1 mL, GE Healthcare). α-syn was eluted with a buffer containing 50 mM Tris–HCl, 100 mM NaCl, and 250 mM imidazole at pH 8.0. The eluted samples were concentrated by centrifugation at $3000 \times g$ for 15 min using Vivaspin Turbo (5 K MW) tubes (15 mL) with buffer containing 50 mM Tris–HCl and 100 mM NaCl, pH 8.0. Proteins were treated with thrombin (GE Healthcare) to remove the N-terminal His-tag. Purified α-syn monomers in 50 mM Tris–HCl containing 100 mM NaCl (pH 8.0) (100 μM, 150 μL) were incubated at 37 °C at 1200 rpm in a shaking incubator (DWMax M/BR-034,Taitec) for 7 days. Measurements at OD 600 (or other wavelengths) were used to check turbidity. α-syn PFFs were pelleted by spinning at $50,000 \times g$ for 20 min and suspended in PBS.

2. Human tau PFFs: Human tau (0N4R recombinant) protein was prepared as described previously[16]. In brief, the homogenate of *Escherichia coli* strain BL21(DE3) expressing human tau in homogenization buffer (50 mM PEPES,1mMEGTA, 1mMDTT, pH6.4) was heated at 95 °C for 5 min and centrifuged at $20,000 \times g$ for 15 min. The resulting supernatant was applied to Cellfine Phosphate (JNC Corp, cat. 19545) and eluted by 1 M NaCl. The tau containing fractions were precipitated by saturated ammonium sulfate, and the pellets were dissolved in 0.05% formic acid. The solution was fractionated by reverse-phase HPLC and purified tau was obtained. For in vitro aggregation, 10 mM recombinant tau proteins in 10 mM HEPES, 100 mM NaCl (pH 7.4) mixed with 60 mg/mL of heparin was incubated at 37 °C at 1200 rpm in a shaking incubator for 7 days.

3. Abeta PFFs: Human $A\beta_{1-42}$ peptides was purchased from Anaspec (Fremont, CA, USA, Cat: AS-24224). The peptides were dissolved in 300 μL $1\%NH_4Cl$ for stock solution and then diluted at 1 mg/mL in 50 mM Tris buffered saline (TBS). The peptides were incubated at 1200 rpm in a shaking incubator for 3 days.

The formation of above PFFs was monitored in 25 μM Thioflavine-T (ThT, SIGMA, MD, USA, cat:T3516) diluted in 50 mM Tris–HCl, 100 mM NaCl (pH 8.0) and at 430 nm excitation and 500 nm emission by Envision multiplate reader (Perkin Elmer, MA, USA). Those PFFs were sonicated by BIORUPTOR (UCD-300, Cosmo Bio Co., Japan) and stored at −80 °C until use.

**Coupling reaction of QD and PFFs.** Carboxyl QD (Cadmium selenium telluride, CdSeTe-COOH, 8.0 μM in borate buffer, cat: Q21361MP), which emits maximum fluorescence intensity at 705 nm, was purchased from Thermo Fisher Scientific (Waltham, MA, USA). Coupling reaction of carboxyl group of QD and PFFs were conducted with Amine Coupling Kit (DOJINDO, Rockville, MD, USA, cat: A515) as shown in Supplemental Fig. S1A. The procedures were conducted according to the manufactured protocols. In brief, the carboxy group of QD was activated by the mixture of reaction buffer (1:1 volume:volume) and incubated with N-hydroxysuccinimide (NHS)/1-(3-dimethylaminopropyl)-3-ethylcarbodiimide (WSC) conjugate solution(QD: reaction buffer: NHS:WSC = 1:1:1:1) (i.e., activated QD mixture). Amino groups of PFFs were activated by activation buffer (PFFs:activation buffer = 1:1) (i.e., activated PFF). After incubation of mixture of activated QD and activated PFFs for overnight at 4 °C, the non-conjugated QDs was blocked by blocking buffer (QD: blocking buffer volume = 1:6). The suspension was spun at $15,000 \times g$ for 10 min. The pellets were resuspended in PBS and checked their fluorescence emission at 705 nm (cy5 filter) for the availability. These QD labeled PFFs were sonicated with BIORUPTOR at ten times of pulses (duration 10 s, interval 10 s), and the PFFs (estimated concentration: QD-α-syn-PFF: 6.3 mg/mL, QD-$A\beta_{1-42}$ -PFF: 1.0 mg/mL, QD-tau-PFF: 5.9 mg/mL) were stored in −80 °C until use.

**Transmission electron microscope (TEM).** Samples for TEM were prepared for suspended three samples: α-syn-seeds (6.3 mg/mL), QD only or QD-α-syn seeds itself. 1 μL of the samples were applied onto 200 mesh uncoated copper grids. The samples were contrasted with 2% uranyl acetate and lead citrate, and the images were recorded with Hitachi HT7700 electron microscopy.

**Thioflavin T (ThT) assay.** In a 96-well plate, 2.5 μL of α-syn seeds, QD and QD-α-syn seeds were mixed with 25 μM ThT in 100 μL of 50 mM Tris–HCl, 100 mM NaCl (pH 8.0) and at 430 nm excitation and 500 nm emission by Envision multiplate reader (Perkin Elmer, MA, USA).

**MTT assay.** MTT assay in neuro2a cells was conducted using CellQuanti-MTT Cell Viability Assay Kit (Cat: CQMT-500, BioAssay Systems, Hayward, CA, USA) according to the manufacturer's protocol. In brief, the neuro2a cells were cultured in Dulbecco modified eagle medium (DMEM, Cat: 08459-35, Gibco) supplemented with 10% fetal bovine serum (FBS, cat: 10270, Gibco). After the cells were stably cultured in $CO_2$ incubator (5% $CO_2$, 37 °C), cells were plated in a 96-well chamber at a concentration of $3 \times 10^4$–$5 \times 10^4$ cells per well. PBS + 0.1% saponin solution (final conc.) was applied as positive control for cytotoxicity. At 24 h after PBS as

control treatment, 10 and 100 μM non-QD-labeled or QD-α-syn seeds were administered, cells were incubated at 37 °C for 1 h. Absorbance of $OD_{570}$ was measured by the plate reader (Envision2104, Perkinelmer, MA, USA).

**Amyloidogenic protein monomer labeling with QD.** The α-syn-monomer was biotinylated and labeled by Qdot 705 streptavidin conjugate. Biotinylation of human α-syn-monomer(200 μg, without tag) was reacted with 11 mM sulfo-NHS Biotin solution (EZ-link Sulfo-NHS-biotin-kit, Cat: 21925, Thermofisher Scientific, IL, USA). After incubation at room temperature for 1 h, non-reacted biotin was removed by Zeba desalt spin column including in EZ-link Sulfo-NHS-biotin-kit. After washing with this column with 1 mL PBS for three times, the suspension including biotinylated monomer and non-reacted biotin were applied to Zeba desalt spin column to remove non-reacted biotin. Because non-reacted biotin was trapped with the column, biotinylated monomers were collected at the bottom of tube. The biotinylated monomer was reacted with Qdot 705 streptavidin conjugate (1 μM, Cat: Q10163MP, Thermofisher Scientific, IL, USA) for 1 h at room temperature. Non-reacted QD was removed and QD-labeled monomers were concentrated by 3 K amicon ultra column (Cat: C82301, Merck Millipore Ltd., Ireland). QD-labeled monomer (estimated concentration: QD-sa-α-syn monomer: 1.0 mg/mL) was stored at −80 °C until use. α-syn-seeds (6.3 mg/mL) were also biotinylated and labeled with Qdot 705 streptavidin conjugate (i.e., QD-sa-α-syn seeds) as wells the as α-syn monomer. The QD-sa-α-syn seeds were applied to imaging in the slice experiment (Supplemental Fig. S5).

**Fluorescent labeling of PFFs with Alexa Fluor 488.** Fluorescent labeling of PFFs were performed with Alexa Fluor 488 carboxylic acid, succinimidyl ester (Alexa Fluor 488 NHS-esters, Cat: A20000, Invitrogen, MA, USA). For labeling with Alexa Fluor 488, the procedure was performed as described elsewhere[17]. In brief, the Alexa-NHS ester was dissolved in D-methyl-surfoxyde (DMSO) at 1.0 mg/mL. PFFs in PBS were incubated with Alexa Fluor 488 NHS-esters (concentration ratio: 1:1) at 4 °C. After incubation, the mixture was spun at $15,000 \times g$ for 10 min to remove the residual unconjugated Alexa and the pellets were washed twice with PBS. Alexa 488 labeled PFFs were sonicated with BIORUPTOR at ten times of pulses (duration 10 s, interval 10 s). The samples were stored at −80 °C until use.

**Injection of fluorescent-labeled PFF seeds to mice.** We purchased 8-week-old male mice (C57BL6J) from Shimizu animal supply (Kyoto, Japan). All of the following experimental procedures for mice were conducted in accordance with the declaration of Helsinki and approved by the ethical committee of Doshisha University. Injection was performed according to the methods described previously[14]. In brief, PFF protein concentrations were determined by BCA assay. The final concentrations of PFFs were set as follows: α-syn seed: 6.3 mg/mL; $A\beta_{1-42}$-PFF, 1.0 mg/mL; tau-PFF: 5.9 mg/mL. Estimated QD or Alexa 488 labeled proteins were similar concentration. 2 μL aliquots (e.g. QD or alexa488 fluorescence labeled PFFs or non-labeled mouse α-syn PFFs) were injected to the right striatum by Hamilton syringe at 0.2 μL/min after drilling the skull ($X = 2.2$ mm, $Y = 0.2$ mm, $Z = 2.5$ mm from bregma) under anesthesia including medetomidine (0.3 mg/kg), midazolam (4.0 mg/kg), and butorphanol (5.0 mg/kg).

**Immunohistochemistry.** Immunohistochemical staining was performed for frozen brain sections and paraffin embedded sections. For frozen sections, thin slices (10 μm thick) were obtained from PFF-seed injected mice. Mice were fixed by perfusion with 4% paraformaldehyde in phosphate buffered saline (PBS, pH 7.2). After the fixation, the brain was submerged in sucrose solution including 30% sucrose, 0.05% $NaN_3$ in 0.1 M phosphate buffer[19]. Prior to immunostaining, slides were incubated in blocking one histo (Nacalai, Kyoto, Japan) for blocking the nonspecific binding of antibodies. Then, slices were incubated with primary antibody at 4 °C, for overnight and secondary antibody at room temperature, for 2 h diluted in dilution buffer including 0.01 M phosphate buffer, 0.5 M NaCl, pH 7.2, 3% bovine serum albumin, 5% normal goat serum, 0.3% Triton-x100 and 0.05% $NaN_3$[19]. Antibodies used were as follows: (1) primary antibodies: anti-α-syn (mouse monoclonal, LB509, 1:100, Invitrogen, MD, USA) for acute α-syn seeds, vesicle-associated membrane protein (vamp) 2 (mouse monoclonal, 104211, 1:100, Synaptic systems Gottingen Germany) for vesicle-associated membrane protein, EEA1 (mouse monoclonal, E41120, 1:100, Transduction laboratories Null USA) for early endosome, LAMP1 (mouse monoclonal, 09671D, 1:100, Pharmingen Franklin Lakes, NJ, USA) for lysosomal-associated membrane protein 1, LAMP2 (mouse monoclonal, 558756, 1:100, Pharmingen) for lysosomal-associated membrane protein 2, LRRK2 (rabbit monoclonal, 3514-1, 1:100, Epitomics, Balingen, CA, USA) for leucine-rich repeat kinase 2, 22C11 (mouse monoclonal, MAB-348, 1:100, Chemicon, Bilerica, MA, USA) for amyloid precursor protein, anti-DARPP32 (rabbit polyclonal, AB10518, 1:100, Millipore, MA, USA), anti-tyrosine hydroxylase (mouse monoclonal, MAB-318, 1:100, Milipore, MA, USA). (2) secondary antibodies: Alexafluor 488 highly cross-absorbed (goat anti-mouse or goat anti-rabbit, A11029 or A11034, 1:100, Invitrogen MD USA). After the immunostaining, slices were incubated with autofluorescence quenching kit (Vector laboratories, Burlingame, CA, USA, cat: SP-8500) to suppress autofluorescence. Recording of frozen section was performed with Keyence microscope (BZ-X710, Keyence, Japan) in optical sectioning mode which is comparable to capture on laser

confocal microscope. For paraffin-embedded sections, immunostaining of brains injected with non-labeled mouse α-syn seeds were performed with the methods described previously[14]. In brief, the autoclaved paraffin sections were incubated with blocking solution containing 5% skim milk in TBST (20 mM Tris–HCl, pH 8.0, 150 mM NaCl, 0.05% Tween 20) for 1 h. Sections were incubated with the primary antibody, phosphorylated α-syn (mouse monoclonal, pSyn#64, 1:300, Wako Japan) in TBST overnight at 4 °C, followed by the secondary antibodies. For diaminobenzidine (DAB) staining, sections were quenched with 3% $H_2O_2$/methanol for 30 min before blocking and incubated with the VECTASTAIN Elite ABC Kit reagent (Vector Laboratories) for 30 min after secondary antibody incubation. To count p-syn inclusions, images of the whole brain sections were recorded by Keyence microscope with a bright field for paraffin sections. Multiple part of fields were captured by a ×10 objective lens and stitched together using the Keyence Merge function. The p-syn deposits per area were quantified by BZ-X710 Generation Analyzer (Keyence).

**Slice preparation**. Acute brain coronal slices (200 μm thick) including striatum, corpus callosum and cortex were obtained and tightly attached on glass bottom dish (IWAKI, Japan) pre-coated with poly-L-lysine (Sigma-Aldrich, St. Louis, USA). The slices were half-submerged with culture medium including Neurobasal-A (Gibco) and horse serum (volume, 5:1) with antibiotics, penicillin and streptomycin (100 μg/mL) . The half-submerged slices on the glass bottom six-well plates (IWAKI, Japan) were maintained in 37 °C, 95% $O_2$/5% $CO_2$ humidified chamber during time lapse imaging.

**Time lapse imaging**. Time lapse imaging was conducted by Keyence fluorescence microscope (BZ-X710). Recording was performed at every 20 s. Dynamics of QD-labeled PFFs seeds were recorded with BZX Cy5 filter (Keyence Co., Osaka, Japan) and exposure time is usually set to 1 s. Objective lens used were (1) Plan Apo ×2 for recording total picture of slices, (2) Plan APO ×4 for recording contralateral to the injected side of slices, and (3) Plan APO ×10 for recording spatio-temporal dynamics of QD-labeled seeds in the regions of interest containing corpus callosum, striatum and cortex. Images of fluorescence and bright field were recorded alternately. Bright field was captured to check the appropriate focusing on slices during each step of time lapse recordings. Recording data, which got out of focus (e.g. due to the drift of slices, etc.), were discarded from data storage. Particle tracking of QD-labeled seeds were conducted by Image-J plug-in software, Trackmate. The rationale was described elsewhere[18]. The parameters were set as follows: select DoG detector, estimated blob diameter: 5, threshold: 1, select hyperstack displayer, select LAP tracker, and parameters used were default. The number and speed of QD-labeled seeds at the region of interest were also analyzed by Trackmate. The number and speed were normalized by the value at the pre-treatment. When the solvent (e.g. dimethyl sulfoxide, ethanol, etc.) was necessary to prepare the dilution of drugs, the same concentration of solvent was used for the pretreatment solution.

**Pharmacological studies**. Drugs were bath-applied in the time lapse imaging experiment. The concentration of these drugs was determined by previous reports. For the time lapse imaging, we prepared the drugs in the stock solution (e.g. PBS or DMSO, etc.) and diluted to the culture medium for recording solution as follows (manufacture, Cat. number, concentration in recording solution, reference describing the appropriate working concentration): dynasore, Tokyo Chemical Industry Co. Japan, D5461, 80 μM[56]; Colchicine, Wako Japan, 035-03853, 10 μM[57]; Botulinum toxin, Biological laboratories Japan, 139, 5.0 nM[14]; gamma-amino-n-butyric acid (GABA), SIGMA, MD, USA, A2129, 1.0 mM[58]; Bicuculline, WAKO, Japan, 026-16131, 20 μM[59]; ampakine, CX-516, SIGMA, MD, USA, SML1191, 30 μM[60]; DNQX, TOCRIS, Bristol, UK, 3693, 40 μM[61]; perampanel, Eisai Co. Ltd, Japan, ER-155055-90, 3 μM[62]; 2-amino-6-trifluoromethoxy-benzothiazole (Riluzole), Tokyo Chemical Industry Co., Japan, A2423, 10 μM[63]; sertraline, Tokyo chemical industry Co. Japan, S0507, 20μM[30]; rifampicin, fuji film Japan, 189-01001, 100 μM[64]; Chilliobrevin D (Chil D), MERCK Millipore, Mam, USA, 250401, 100 μM[65]. S-trityl-L-cystein (STLC), Abcam, Cambridge, UK, ab144578, 50 μM[66]. First, pre-recordings (i.e., control without drugs) were performed, then, 30 min after the administration of drugs, imaging for drug effects was recorded. After the recording for drug effects, we replaced the medium with dissolving solvent (e.g., DMSO) and tested the washout for recovery.

For studies of in vivo effects, we carried out intraperitoneal injection of drugs every day from 7 days before the injection of seeds until the sacrifice of the mice. The concentrations used were as follows: riluzole, 20 mg/kg[67]; perampanel, 20 mg/kg[29]; sertraline, 20 mg/kg[68]; and rifampicin, 20 mg/kg[31].

**Western blotting**. Brain tissues of the ipsilateral hemisphere (i.e., side of amyloids) and contralateral hemisphere at 2 months after injection of mouse α-syn were dissected into 2 mm-thick coronal slices, which included cortico-striatum. Slices were homogenized with disposable homogenizer SP and power masher II (Nippi, Inc., Tokyo, Japan). For sarkosyl soluble and insoluble fractions, the tissue homogenates were diluted in buffer (final; 10 mM Tris–HCl, pH 7.6, 0.8 M NaCl, 10% sucrose, 0.1% triton x-100, 50 mM NaF, 1 mM $Na_3VO_4$ + cOmplete (Sigma-aldrich, 1 tablet/50 mL) as well as described previously[69]. 2% 2-mercaptoethanol,

20% glycerol, and 0.01% bromophenol blue were added to make final samples (protein concentration; 0.7 mg/mL). Samples (30 μg) after boiling were loaded onto 12% SDS containing polyacrylamide gels (6.25% stacking gel) and separated at 20 mA for 80 min at room temperature with running buffer (100 mM Tris, 100 mM glycine, and 0.1% SDS). Proteins in the gels were transferred onto a PVDF membrane for 90 min at 115 mA at room temperature. The membranes were blocked in 5% skim milk in 0.05% Tween 20/Tris-buffered saline (TBST) and incubated with anti-p-syn antibody (mouse monoclonal, pSyn#64, 1:500, Wako, Japan) overnight at 4 °C. The membranes were washed three times in TBST and incubated for 1 h with horseradish peroxidase-conjugated secondary antibody (dilution 1:2000). Immunoreactive proteins were visualized using a chemiluminescence reagent, Luminata (Millipore). Chemiluminescent signals were obtained using ImageQuant LAS-4000 (GE Healthcare).

**Filter trap assay**. Filter trap assay for detection the amount of insoluble high molecular weight α-syn aggregates were conducted using a HYBRI-DOT MANIFOLD (Cat: 1050MM, BRL Life Technologies Inc.) and cellulose acetate membrane filter with a pore size of 0.2μm (Cat: C020A142C, Advantec) as described previously[69]. In brief, the samples (5 μL) were mixed in 2% SDS and PBS and applied to apparatus. Soluble proteins were removed by vacuum suction and the SDS-resistant aggregates were trapped at the membrane filter. The filter was washed three times with 2%SDS/PBS solution, and suction was maintained for 20 min. The membrane was subsequently blocked by 5% skim milk. Immunostaining of the filter and detection of signals were performed the same as western blot.

**Tubulin polymerization and labeling with QD**. Formation of microtubule was performed using HTS-Tubulin Polymerization Assay Biochem Kit (Cat: BK004P, Cytoskeleton, Inc., Denver, CO, USA) according to the manufacturer's protocol. In brief, porcine tubulin monomer (including in kit) was mixed with tubulin glycerol buffer (5% glycerol), 1 mM general tubulin buffer (80 mM PIPES, pH = 6.9, 2 mM $MgCl_2$, 0.5 mM EDTA) and 5% GTP. The mixture solution was incubated in 37 °C and measured $OD_{340nm}$ by SpectraMax M2 (Molecular Devices, CA, USA) for 1 h. The parameter settings were set as follows: (1) measurement type: kinetic, 120 cycles of 1 reading per 30 s, (2) absorbance wavelength: 340 nm, (3) temperature: 37 °C, (4) shaking: once at start of reaction, 5 s medium, orbital. The solution was collected after the reaction, and the concentration was determined by BCA assay kit (Pierce Biotechnology, MA, USA). The microtubules were collected by centrifugation at $100,000 \times g$ for 45 min and the pellet was treated by 5% glutaraldehyde for fixation. For the western blot, anti-β-tubulin (Cat: MAB1637, Chemicon) was used. The microtubules and monomer were labeled with QD by the same method for QD-labeling of PFF. The QD-labeled microtubules were sonicated with BIORUPTOR as QD-labeled PFF were treated. The QD labeled monomer and microtubules were injected to mouse striatum as described for QD-α-syn seed injection and the brains were processed for slice imaging.

**Statistics and reproducibility**. Results were expressed as mean ± SEM. Comparison of two sets of data was analyzed by Student's $t$ test. Multiple comparisons were conducted by one-way ANOVA followed by Tukey's post test. $p$ value < 0.05 was evaluated as statistically significant. Data were analyzed by GraphPad Prism 7.0 software (SanDiego, CA, USA).

**Reporting summary**. Further information on research design is available in the Nature Research Reporting Summary linked to this article.

## Data availability

The dataset used in this study are available on https://doi.org/10.6084/m9.figshare.19727125.v2. These Source data files are provided as Supplemental Data 1–11 and Supplemental Movies 1–4.

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

## Acknowledgements

The authors thank Ce Xie for the initial preparation of recombinant tau and Dr. Soichiro Kakuta, the Laboratory of Morphology and Image Analysis, Research Support Center, Juntendo University Graduate School of Medicine for technical assistance with electron microscopy. We appreciate Eisai Co., Ltd. for providing perampanel used in this study. This work was supported by Japan Agency for Medical Research and Development, AMED under Grant Number JP20dm0107140 to N.N., and from the Ministry of Education, Culture, Sports, Science and Technology (MEXT) of Japan to N.N. (17H01564) and Y.I. (20K21587).

## Author contributions

Y.I. did all of the experiments. A.O. and N.H. participated in the immunohistochemical and TEM analysis. S.Y. and A.H. performed the preparation of amyloidogenic proteins with the help of T.M. Y.F. performed structural analysis of fibrils. N.N. supervised this project and completed this manuscript with Y.I.

## Competing interests

The authors declare no competing interests.
