## [Peer Review File · Communications Biology]

Reviewers' comments:

Reviewer #1 (Remarks to the Author):

The authors investigated the incorporation and intracellular transport of labeled alpha-synuclein preformed fibrils (PFF) (and also other amyloid fibrils such as fibrils made of Amyloid beta peptide and tau protein) in slices of mouse brain tissues, and demonstrated that the fibrils are mainly transported retrogradely in corpus callosum and blocked by a dynein inhibitor (Chil D) by showing the movies. The incorporation of PFF and the transport were also affected by various drugs, such as dynasore (endocytosis inhibitor), colchicine (microtubule depolymerization), botulinum toxin (inhibitor of vesicle release), an anterograde transport inhibitor, etc. GABA agonist (GABA), AMPAR antagonist (DNQX), and NMDAR antagonist also significantly reduced the migration number of a-syn seeds. Finally, the authors tested some clinically available drugs (riluzole, perampanel, sertraline and rifampicin) on a-syn seed dynamics in vitro and in vivo, and demonstrated the pretreatment of these drugs reduced the formation of phosphorylated alpha-synuclein pathology.

This is a very interesting study to think about the molecular mechanisms of prion-like propagation of amyloid-like intracellular proteins. It is also interesting to note that the drugs that were effective on the a-syn pathologies in this paper may have potential for therapeutic applications.

Major comments.

1. Authors used various drugs in this study. It would be helpful to the reader if the authors summarized the efficacy of each drugs and the effect obtained in this study in a table in both their slice experiments and in vivo experiments.
2. In this paper, all amyloid or amyloid-like fibrils seem to be taken up and transported in the same way. Authors tested and described that the QD only were not incorporated and transported. However, the authors should test the QD-labeled soluble a-syn monomer as the control. It will be interesting to know if soluble monomer do or do not behave like the fibrils.
3. It is also interesting to know whether physiological neurofilaments and/or fixed microtubules behave similarly to those observed in amyloids.
4. In the title " Quantum-dot-labeled synuclein seeds" gives the impression that it is better than the use of fluorescent dye labeled proteins. However, in their supplementary data the Alexa488 labeled seeds gave the same or even better results. Please discuss about the necessity of using QD labels in more detail, how different they are or else I think you don't need to include it in the title.
5. The axonal transport of the fibrils in the movies are very fast and very impressive. Is this high-speed transport limited in the corpus callosum or is it same as in the other regions?
The reviewer would like to ask you to discuss if it is as fast as normal retrograde transport of normal protein or organelle transport.
6. In vivo effect of the drugs should be examined whether the drug is effective when it is administered after seed inoculation. Authors should treat the mice with drugs at 1~2 weeks after inoculation. If there is no effect, it can be understood that the drugs may work on the initial seed uptake.

Minor point

Regarding the relationship between the neuronal activity and a-syn propagation, Wu et al reported that neuronal hyperactivity promoted PFF trafficking along axons/dendrites within microfluidic chambers in *Acta Neuropathologica* 2020. Authors should cite the paper.

Reviewer #2 (Remarks to the Author):

Review NC "Quantum-dot-labeled Synuclein Seeds Assay Identifies Drugs Modulating Prion-line Transmission"

Overview: In this manuscript the authors investigate the mechanism of recombinant synuclein PFF seeding and spreading in acute slices of brain tissue and present data suggesting the PFF seed spreading is dependent on endocytosis and neuronal activity while the trafficking is dependent on fast axonal transport. This feat is accomplished using quantum-dot labelled labeled PFFs to directly visualize uptake and spreading of synuclein aggregates in vivo. Additionally this assay was used to identify drugs that are capable of inhibiting these mechanisms and some that might be used

therapeutically. Currently this is a very active field of research with great interest in understanding the toxic species of synucleinopathies and the mechanisms of disease progression. This paper demonstrates the benefit of using QD-labeled seeds to address numerous mechanistic questions, but there are several ways it should be improved. First any paper discussing a prion-like mechanism must demonstrate that filaments are formed upon seeding, and this is especially true when presenting a novel assay. Second, these experiments should be done using patient tissue extracted pathogenic protein aggregates to demonstrate disease relevance.

Recommendation:

I recommend this manuscript for publication with significant revisions.

Major Revisions:

This manuscript clearly demonstrates the functionality and utility of using QD-labeled seeds to track protein spread throughout the brain. The importance of this manuscript would be improved dramatically by using extracted pathogenic seeds from patient tissues instead of recombinant synuclein PFFS which are not reported to be structurally related to disease associated synuclein filaments. This would improve the relevance of this paper significantly for future translational work. Additionally and technically relevant to the manuscript presented, data needs to be presented clearly demonstrating that aggregated filamentous synuclein is observed on the contralateral side after seed injection. This is required to validate the new assay and begin demonstrating that prion-like seeding is in fact occurring. If filaments are not known to be formed after seeding than it is not possible to know that the seeding is leading to prion-like templating and spreading.

Minor Revisions:

Title Page

Page 1

Lines 1-2: The title is awkward. Consider something more direct...“QDL-labeled synuclein seeds used to visualize prion-like transmission and identify potential therapeutics”

Introduction

Page 4

Lines 46, 50, 56: Replace a-syn with alpha-syn. Check throughout text.

Results

Page 6

Lines 87-88: Supplemental Fig S1b TEM micrographs are not clear. The filaments do not look labeled. There is only an amorphous QD-syn seed aggregate shown. Please correct. Add before/after sonication.

Line 92: It is quite interesting to see the seeds on the contralateral side after 1 hour. Was this confirmed to not be an artifact in a previous paper perhaps by lesioning the CC and then later seeding? If so please provide a reference and/or highlight again.

Page 7

Lines 99-100: It looks like from the images provided and quantitation that both the iSt and cSt both show significant colocalization.

Lines 97-110: Please indicate which negative controls were run. Were QD-labeled non-pathogenic proteins or monomeric synuclein injected as controls? Please indicate the controls in the text.

Lines 111-114: Incorporate contralateral data above.

Page 8

Lines 127-128: Again indicate if unassembled synuclein or another unassembled protein was run as a negative control.

Page 9

Lines 133-150: It says in the methods seed numbers and seed rates were tracked for the treatments. Please include both sets of data for each treatment in the text. What about combined treatments.

Page 10

Lines 155-164: Please include both the seed number and velocity data in the text for these treatments as well. Or simply reference the figure. Just be consistent.

Page 12

Lines 195-203: Please indicate if biochemistry and TEM of the contralateral sides of the brains was performed to show reduced aggregated synuclein as indicated by the IHC. There is no data on the nature of these aggregates, but it needs to be shown they are filamentous for prion-like seeding studies.

Page 13

Lines 206-216: These studies should all be performed with QD-labeled extracted pathological filamentous aggregates that are directly disease relevant.

Discussion

Page 13

Lines 219-222: The first sentence of the discussion indicates that it has to be shown that the seeded aggregates are in fact synuclein protein filaments for a prion-like mechanism.

Reviewer #3 (Remarks to the Author):

The manuscript by Imamura et al. present versatility of 'quantum dots-labeled α -Syn seeds' in understanding the mechanistic pathways involved in trafficking and prion-like transmission of α -Syn. Authors have extensively studied the movement of α -Syn seeds using various pharmacological inhibitors. Authors further demonstrated tracking of QD-labeled α -Syn seeds over time, and provided evidence for the involvement of axonal transport and neural activity for early phase spreading of α -Syn seeds. Authors have further tested various pharmacological drugs viz. Riluzole, perampanel, sertraline etc., on studying migration and spreading of α -Syn seed pathology in vivo. In summary, the manuscript provide insights into an effective drug screening platform not only for synucleinopathies but also for other amyloid disorders.

Overall, the study is well designed and the results are supported by the provided data. The manuscript is well-written and cites the relevant literature in this field. The reviewer is of the opinion that the manuscript can be accepted for publication in Communication Biology after addressing the following comments and suggestions

- a. TEM images presented in supplementary figure 1b are low resolution images. Authors needs to replace the figure with high resolution images.
- b. Author should provide CD spectra or FTIR spectra to show β -sheet structure of α -Syn fibrils, which is used for in vivo experiments.
- c. Multiple reports suggests that, quantum dots inhibit fibrillation of α -Syn or cause disaggregation of α -Syn (Kim et al.,2018; Ghaeidamini et al.,2020). Therefore, author should provide the CD/FTIR spectra and ThT binding data for understanding the effect of quantum-dot labeling on secondary structure of α -Syn fibrils.
- d. Author should check the cytotoxicity of QD labeled- α Syn seed in neuronal cells and compared with equimolar concentration of naive α -Syn seed.
- e. Author should provide magnified images for immune staining (p-syn immunoreactivity) for Fig 5g and 5h.
- f. Colocalization of QD- α -syn-seeds fluorescence and α -Syn immunoreactivity have been shown in supplementary figure 1c. However, author should include the quantification of the colocalized signal between QD- α -syn-seeds and LB509 immunoreactivity.
- g. Authors have used various pharmacological drugs for understanding the mechanistic characteristics of spreading of α -Syn pathology in vivo. However, it is not clear that, how authors had optimized these drug concentration? Is it based on any preliminary studies? It should be discussed in methodology part.
- h. Schematics in Fig 6 is not clear. Author may include an elaborated schematic illustration specifying the overall outcomes of the study.

Response to reviewers

We appreciate the reviewers' constructive and helpful comments and revised our manuscript according to their comments (point by point response).

Reviewer #1 (Remarks to the Author):

The authors investigated the incorporation and intracellular transport of labeled alpha-synuclein preformed fibrils (PFF) (and also other amyloid fibrils such as fibrils made of Amyloid beta peptide and tau protein) in slices of mouse brain tissues, and demonstrated that the fibrils are mainly transported retrogradely in corpus callosum and blocked by a dynein inhibitor (Chil D) by showing the movies. The incorporation of PFF and the transport were also affected by various drugs, such as dynasore (endocytosis inhibitor), colchicine (microtubule depolymerization), botulinum toxin (inhibitor of vesicle release), an anterograde transport inhibitor, etc. GABA agonist (GABA), AMPAR antagonist (DNQX), and NMDAR antagonist also significantly reduced the migration number of a-syn seeds. Finally, the authors tested some clinically available drugs (riluzole, perampanel, sertraline and rifampicin) on a-syn seed dynamics in vitro and in vivo, and demonstrated the pretreatment of these drugs reduced the formation of phosphorylated alpha-synuclein pathology.

This is a very interesting study to think about the molecular mechanisms of prion-like propagation of amyloid-like intracellular proteins. It is also interesting to note that the drugs that were effective on the a-syn pathologies in this paper may have potential for therapeutic applications.

Major comments.

1. Authors used various drugs in this study. It would be helpful to the reader if the authors summarized the efficacy of each drugs and the effect obtained in this study in a table in both their slice experiments and in vivo experiments.

Response to Reviewer 1 Q1: The effects of drugs *in vitro* and *in vivo* are summarized in Table 1, which has been added to the revised manuscript.

2. In this paper, all amyloid or amyloid-like fibrils seem to be taken up and transported in the same way. Authors tested and described that the QD only were not incorporated and transported. However, the authors should test the QD-labeled soluble a-syn monomer as the control. It will be interesting to know if soluble monomer do or do not behave like the fibrils.

Response to Reviewer 1 Q2: Because our labeling method oligomerized the monomers due to multiple reaction sites of QD (the labeled oligomers could not disseminate to the contralateral side, data not shown), we labeled α -syn monomer with QD and QD- α -syn seeds by the biotin-streptoavidin method and tested their behavior. QD- α -syn monomers were not observed in the ROI (Suppl. Fig S3, Suppl. Fig S5) and the labeled fibrils were moving in the ROI as the QD α -syn fibrils labeled by our methods. Results are shown in supplemental Fig. S5.

3. It is also interesting to know whether physiological neurofilaments and/or fixed microtubules behave similarly to those observed in amyloids.

Response to Reviewer 1 Q3: It is an interesting and important question. We added an experimental result using QD labeled fixed microtubules in supplemental Fig. S12. QD labeled monomeric tubulin and fixed microtubules were observed significantly less than QD-labeled alpha synuclein fibrils in the ROI, suggesting the enhanced incorporation of amyloid fibrils into the cells. This is discussed in the manuscript.

4. In the title “Quantum-dot-labeled synuclein seeds” gives the impression that it is better than the use of fluorescent dye labeled proteins. However, in their supplementary data the Alexa488 labeled seeds gave the same or even better results. Please discuss about the necessity of using QD labels in more detail, how different they are or else I think you don’t need to include it in the title.

Response to Reviewer 1 Q4: Alexa-488 labeled a-syn-PFF could be seen in this thin frozen section. However, it is hard to observe the movement in thick slices. QD is much brighter than Alexa and shows no photobleaching. Therefore, labeling by QD is very important to track the movement of amyloid-PFF. We added this reason for using QD in the manuscript.

5. The axonal transport of the fibrils in the movies are very fast and very impressive. Is this high-speed transport limited in the corpus callosum or is it same as in the other regions?

The reviewer would like to ask you to discuss if it is as fast as normal retrograde transport of normal protein or organelle transport.

Response to Reviewer 1 Q5: The shown movie is 10X speed. We chose three regions of interest: 1) contralateral cortex, 2) corpus callosum and 3) contralateral striatum. Compared to the corpus callosum (CC), the speed was slower in cortex and striatum (Fig 2d). In the CC, the seeds move in the axons. In the cortex and the striatum, the seeds might move in the proximal axon, cell body or dendrites. The difference of velocity among the regions could be due to the localizations of seeds. This result was added in the manuscript.

6. In vivo effect of the drugs should be examined whether the drug is effective when it is

administered after seed inoculation. Authors should treat the mice with drugs at 1~2 weeks after inoculation. If there is no effect, it can be understood that the drugs may work on the initial seed uptake.

Response to Reviewer 1 Q6: We added an experiment of treating the mice with Riluzole at 2 weeks after seed inoculation. There was no effect on the p-syn pathology (Suppl. Fig. S7). These findings suggest that the drug works on the initial seed uptake. This result was added in the manuscript.

Minor point

Regarding the relationship between the neuronal activity and a-syn propagation, Wu et al reported that neuronal hyperactivity promoted PFF trafficking along axons/dendrites within microfluidic chambers in *Acta Neuropathologica* 2020. Authors should cite the paper.

Response to Reviewer 1 minor point: This paper is now referenced in the introduction of the revised manuscript.

Reviewer #2 (Remarks to the Author):

Review NC “Quantum-dot-labeled Synuclein Seeds Assay Identifies Drugs Modulating Prion-line Transmission”

Overview: In this manuscript the authors investigate the mechanism of recombinant synuclein PFF seeding and spreading in acute slices of brain tissue and present data suggesting the PFF seed spreading is dependent on endocytosis and neuronal activity while the trafficking is dependent on fast axonal transport. This feat is accomplished using quantum-dot labelled labeled PFFs to directly visualize uptake and spreading of synuclein aggregates in vivo. Additionally this assay was used to identify drugs that are capable of inhibiting these mechanisms and some that might be used therapeutically. Currently this is a very active field of research with great interest in understanding the toxic species of synucleinopathies and the mechanisms of disease progression. This paper demonstrates the benefit of using QD-labeled seeds to address numerous mechanistic questions, but there are several ways it should be improved. First any paper discussing a prion-like mechanism must demonstrate that filaments are formed upon seeding, and this is especially true when presenting a novel assay. Second, these experiments should be done using patient tissue extracted pathogenic protein aggregates to demonstrate disease relevance.

Recommendation:

I recommend this manuscript for publication with significant revisions.

Major Revisions:

This manuscript clearly demonstrates the functionality and utility of using QD-labeled seeds to track protein spread throughout the brain. The importance of this manuscript would be improved dramatically by using extracted pathogenic seeds from patient tissues instead of recombinant synuclein PFFS which are not reported to be structurally related to disease associated synuclein filaments. This would improve the relevance of this paper significantly for future translational work. Additionally and technically relevant to the manuscript presented, data needs to be presented clearly demonstrating that aggregated filamentous synuclein is observed on the contralateral side after seed injection. This is required to validate the new assay and begin demonstrating that prion-like seeding is in fact occurring. If filaments are not known to be formed after seeding than it is not possible to know that the seeding is leading to prion-like templating and spreading.

Response to Reviewer 2 Major Points: Regarding the extracted pathogenic seeds, in this study we focused on the experimental prion-like transmission, which has recently been used in many studies and revealed the regulatory mechanism of this experiment. This is the most important point of our study. We are also interested in transmission in the context of human disease, however, using human material is problematic because the quality of the diseased tissue varies widely across cases. For example, in synucleinopathy like diffuse Lewy body disease, Alzheimer pathology is often included to various degrees. In addition, aged brains typically have lipofuscin in the insoluble fraction. Thus, we think using human materials is not ideally suited for the assay system. We believe that for the drugs discovered by our assay system, it is better to observe their effects on transmission with extracted pathogenic seeds from disease brain to create a foundation for future translational work. We discuss the importance of the extracted pathogenic seeds experiment for future translational research in the discussion. Regarding the filamentous aggregates, we did not succeed in finding them in the transmitted area by electron micrography at present, possibly due to a technical problem. As far as we know, many reports using this transmission experiment did not confirm the filamentous aggregates but confirmed the existence of phosphorylated a-syn aggregates (Suppl. Fig. S8d). However, a filter trap assay, which is often used in the study of insoluble proteins, revealed insoluble phosphorylated synuclein aggregates, suggesting that exogenous seeds convert endogenous a-syn to aggregates. In any case, the most important point of our study was to reveal the regulatory mechanism of well-established and frequently used prion-like transmission experiments.

Minor Revisions:

Title Page

Page 1

Lines 1-2: The title is awkward. Consider something more direct... "QDL-labeled synuclein seeds

used to visualize prion-like transmission and identify potential therapeutics”

Response to Reviewer 2: Yes, it may seem awkward, as we did not want to overstate. We would like to change the title to “Quantum-dot-labeled synuclein seed assay identifies drugs modulating experimental prion-like transmission”

Introduction

Page 4

Lines 46, 50, 56: Replace a-syn with alpha-syn. Check throughout text.

Response to Reviewer 2: According to the reviewer’s comment, we changed ‘a-syn’ to ‘ α -syn’.

Results

Page 6

Lines 87-88: Supplemental Fig S1b TEM micrographs are not clear. The filaments do not look labeled. There is only an amorphous QD-syn seed aggregate shown. Please correct. Add before/after sonication.

Response to Reviewer 2: TEM was performed again, and we replaced the images in Supplemental Fig. 1b.

Line 92: It is quite interesting to see the seeds on the contralateral side after 1 hour. Was this confirmed to not be an artifact in a previous paper perhaps by lesioning the CC and then later seeding? If so please provide a reference and/or highlight again.

Response to Reviewer 2: Line 92 in Suppl Fig.1d shows the QD labeled alpha synuclein seeds one hour after injection. As the reviewer kindly states, it confirmed our previous study using callosotomy that the seed rapidly disseminated to the contralateral side. We commented on it and referred to our previous work.

Page 7

Lines 99-100: It looks like from the images provided and quantitation that both the iSt and cSt both show significant colocalization.

Response to Reviewer 2: QD- α -syn seeds seems to be colocalized relatively higher in ipsilateral cortex. However, as the reviewer suggested, a statistically significant difference was not observed among regions. We corrected the description in the revised manuscript.

Lines 97-110: Please indicate which negative controls were run. Were QD-labeled non-pathogenic proteins or monomeric synuclein injected as controls? Please indicate the controls in the text.

Response to Reviewer 2: In this study, we examined the colocalization of organelle markers

with QD-positive seeds (Suppl Fig S3). Because injected QD-labeled monomeric synuclein was not observed in those ROIs, the colocalization with organelle markers was not observed. We described this in the manuscript.

Lines 111-114: Incorporate contralateral data above.

Response to Reviewer 2: The results of QD-labeled monomer injected were added to Figure1 (h, i, l, m) of the revised manuscript.

Page 8

Lines 127-128: Again indicate if unassembled synuclein or another unassembled protein was run as a negative control.

Response to Reviewer 2: We had some difficulties in labeling the monomer by our labeling methods because QD has multiple reaction sites and make oligomers, which, however, could not be observed in the ROI, suggesting those are not endocytosed into the neuron (data not shown). Thus, we labeled a-syn monomer and fibrils with QD using biotin and streptoavidin binding(for monomer, Suppl Fig S5a). QD-sa-a-syn seeds labeled with this method were successfully moving into the contralateral hemisphere(Suppl Fig S5c, lower panel), but the QD-sa-a-syn monomers were not (Suppl Fig S5c, upper panel). Results were shown in Supplemental Fig. S5.

Page 9

Lines 133-150: It says in the methods seed numbers and seed rates were tracked for the treatments. Please include both sets of data for each treatment in the text. What about combined treatments.

Response to Reviewer 2: The seed numbers were added to the revised manuscript. We did not perform the combined treatment, because we would like to know the specific effect of each of those drugs.

Page 10

Lines 155-164: Please include both the seed number and velocity data in the text for these treatments as well. Or simply reference the figure. Just be consistent.

Response to Reviewer 2: The seed numbers and velocity were added to the revised manuscript.

Page 12

Lines 195-203: Please indicate if biochemistry and TEM of the contralateral sides of the brains was performed to show reduced aggregated synuclein as indicated by the IHC. There is no data on the nature of these aggregates, but it needs to be shown they are filamentous for prion-like seeding

studies.

Response to Reviewer 2:

Regarding the nature of these aggregates, we presented the following data. 1) Biochemical data. Western blot and filter trap assay of seeded mouse brain were presented (Suppl. Fig S8). 2) We also presented the phosphorylated a-syn pathology of QD-labeled a-syn seeds to show the seeding property is preserved after QD labelling (Suppl Fig. S9).

Page 13

Lines 206-216: These studies should all be performed with QD-labeled extracted pathological filamentous aggregates that are directly disease relevant.

Response to Reviewer 2: As described above in our response to Major Points, in this study we focused on the experimental prion-like transmission, which has recently been used in many studies and revealed the regulatory mechanism of this experiment. This is the most important point of our study. We discuss the importance of extracted pathogenic seed experiments for future translational research in the revised Discussion.

Discussion

Page 13

Lines 219-222: The first sentence of the discussion indicates that it has to be shown that the seeded aggregates are in fact synuclein protein filaments for a prion-like mechanism.

Response to Reviewer2: We cited previous papers and described this more clearly.

Reviewer #3 (Remarks to the Author):

The manuscript by Imamura et al. present versatility of ‘quantum dots-labeled α -Syn seeds’ in understanding the mechanistic pathways involved in trafficking and prion-like transmission of α -Syn. Authors have extensively studied the movement of α -Syn seeds using various pharmacological inhibitors. Authors further demonstrated tracking of QD-labeled α -Syn seeds over time, and provided evidence for the involvement of axonal transport and neural activity for early phase spreading of α -Syn seeds. Authors have further tested various pharmacological drugs viz. Riluzole, perampanel, sertraline etc., on studying migration and spreading of α -Syn seed pathology in vivo. In summary, the manuscript provide insights into an effective drug screening platform not only for synucleinopathies but also for other amyloid disorders.

Overall, the study is well designed and the results are supported by the provided data. The manuscript is well-written and cites the relevant literature in this field. The reviewer is of the opinion that the manuscript can be accepted for publication in Communication Biology after addressing the

following comments and suggestions

a. TEM images presented in supplementary figure 1b are low resolution images. Authors needs to replace the figure with high resolution images.

Response to Reviewer 3a: TEM experiments were performed again and higher-resolution TEM images were used to replace those in Supplementary Fig. 1b.

b. Author should provide CD spectra or FTIR spectra to show β -sheet structure of α -Syn fibrils, which is used for in vivo experiments.

Response to Reviewer 3b: According to your suggestion, we performed the additional experiments using FT-IR. The result is shown in Figure for reviewer 3. We examined QD only, a-syn-seeds, QD-labeled a-syn-seeds with FT-IR and successfully confirmed the β -sheet structures of α -Syn fibrils that are characterized with absorption at 1620 cm⁻¹. Unfortunately, however, QD itself showed significant absorption at 1620cm⁻¹ (Fig. for reviewer 3a), which makes it hard to distinguish potential effects of QD labelling on the β -sheet structure of α -Syn fibrils. Also, QD has significant absorption in the ultra-violet region; therefore, CD spectroscopy is difficult to apply for characterization of the secondary structures of the protein fibrils. However, the filament structure was not likely to be affected by QD labeling to alpha-syn as shown in Supplementary Fig. 1b.

Figure for reviewer 3 FT-IR

a-c no difference for FT-IR of QD only (a), QD- α -syn seeds(b) and QD and α -syn seeds without conjugation. Please note that QD only showed high values of spectrum at 1620cm⁻¹ which was corresponding to β -sheet structure. It's hard to determine the effect of QD on β -sheet structure by FT-IR spectrum.

c. Multiple reports suggests that, quantum dots inhibit fibrillation of α -Syn or cause disaggregation of α -Syn (Kim et al.,2018; Ghaeidamini et al.,2020). Therefore, author should provide the CD/FTIR spectra and ThT binding data for understanding the effect of quantum-dot labeling on secondary structure of α -Syn fibrils.

Response to Reviewer 3c: As described above, both FT-IR and CD spectroscopy are difficult to characterize the secondary structures of QD labeled α -syn fibrils. Instead, in order to confirm the potency of the QD-labeled fibrils as seeds, we performed ThT assay for non-labeled α -syn fibrils and QD- α -syn fibrils (Suppl Fig S1c). Result indicated that QD- α -syn-fibrils showed higher ThT fluorescence as well as ThT at non-labeled α -syn fibrils.

d. Author should check the cytotoxicity of QD labeled- α Syn seed in neuronal cells and compared with equimolar concentration of naive α -Syn seed.

Response to Reviewer 3d: We checked the cytotoxicity in neuro2a cells with MTT assay. Results are shown in Suppl. Fig. S2 . No significant difference was observed among PBS, α -syn

seeds and QD- α -syn seeds.

e. Author should provide magnified images for immune staining (p-syn immunoreactivity) for Fig 5g and 5h.

Response to Reviewer 3e: Magnified images are provided in Suppl. Fig. S6.

f. Colocalization of QD- α -syn-seeds fluorescence and α -Syn immunoreactivity have been shown in supplementary figure 1c. However, author should include the quantification of the colocalized signal between QD- α -syn-seeds and LB509 immunoreactivity.

Response to Reviewer 3f: The quantification of the colocalized signal between QD- α -syn-seeds and LB509 immunoreactivity is included in revised Suppl. Fig S1e.

g. Authors have used various pharmacological drugs for understanding the mechanistic characteristics of spreading of α -Syn pathology in vivo. However, it is not clear that, how authors had optimized these drug concentration? Is it based on any preliminary studies? It should be discussed in methodology part.

Response to Reviewer 3g: The drug concentration was determined by previous reports. We added citations of previous reports in the Methods of the revised manuscript.

h. Schematics in Fig 6 is not clear. Author may include an elaborated schematic illustration specifying the overall outcomes of the study.

Response to Reviewer 3g: We modified the figure and the legend to explain our result more clearly.

Reviewers' comments:

Reviewer #1 (Remarks to the Author):

I think the authors have adequately addressed the points raised. No further comments from me.

Reviewer #2 (Remarks to the Author):

The presented data was very solid, but it does not address this reviewer's major concerns adequately. Constructive comments were added in the attached review.

Reviewer #3 (Remarks to the Author):

The authors have satisfactorily addressed the reviewer's concerns and the revision has substantially improved the manuscript. The findings of this study seem promising and have a significant impact in understanding the uptake, and prion-like transmission of α -Syn seeds. The versatility of this technique may be used as a drug screening platform for synucleinopathies. Therefore, I recommend that the manuscript can be published in 'Communication Biology' in the present form.

Responses to the reviewers' comments

We appreciate the reviewers' comments to our revised manuscript.

We carefully checked the reviews for our revised manuscript.

Two reviewers were satisfied with our revised manuscript. However, the reviewer #2 was not.

This reviewer requested two things: 1) Detection of filaments formed upon seeding, 2)

Experiment using patient tissue extracted pathogenic protein aggregates.

I would like to respond to these comments.

This reviewer said at the first review that the title of our paper should be changed to "QDL-labeled synuclein seeds used to visualize prion-like transmission and identify potential therapeutics." And we answered that as we did not want to overstate. We would like to change the title to "Quantum-dot-labeled synuclein seed assay identifies drugs modulating experimental prion-like transmission". Also in the second review, this reviewer said "If this is strictly a seeding uptake paper, then this part should be expanded to include other pathogenic proteinaceous seeds associated with neurodegeneration". Based on this discussion, it seems that this reviewer applied his or her own standards or criteria of seeding

to our manuscript: when the manuscript discusses about the prion-like transmission, filaments should be demonstrated and when it discusses about the therapy, patient materials should be used. However, we think scientific paper should focus on the subject clearly. We focused on the in vivo transmission experiment, which is recently used in many papers in the prion-like transmission field and revealed the mechanism of transmission in this type of experiments. This is very important in the field and two other reviewers reasonably understood the significance of our paper.

I would like to respond to two points raised by this reviewer #2.

- 1) Using the in vitro experiments, it has been easily demonstrated that filaments seeds induce filaments, however, in vivo transmission experiments, accumulation of abnormal phosphorylated α -synuclein were demonstrated, but filaments were not demonstrated in situ (in the transmitted animal specimen) by the electron microscopy (EM) even in the following high impacted papers dealing the prion-like transmission (1-7). Based on our experience, I think this is because in vivo condition is different from in vitro and the environments in vivo could reduce the fibrillar formations and oligomer or prefibrils are formed, or much longer time, such as several years in human disease, is necessary to form large fibrils, which could be detected by EM. Anyway, the filament formation detected by

EM is large and even the filaments are not detected, the molecular structure could be replicated. Thus, those important papers (1-7) did not show the filaments in situ. Anyway, we are focusing on the prion-like transmission experiment, which is used in many other papers and on the regulatory mechanism of this experiment. We do not want to discuss about the validity of this experiment as this reviewer did.

References

- 1 Watts, J. C. *et al.* Transmission of multiple system atrophy prions to transgenic mice. *Proc Natl Acad Sci U S A* **110**, 19555-19560, doi:10.1073/pnas.1318268110 (2013).
- 2 Peelaerts, W. *et al.* alpha-Synuclein strains cause distinct synucleinopathies after local and systemic administration. *Nature* **522**, 340-344, doi:10.1038/nature14547 (2015).
- 3 Rutherford, N. J. *et al.* Comparison of the in vivo induction and transmission of alpha-synuclein pathology by mutant alpha-synuclein fibril seeds in transgenic mice. *Hum Mol Genet* **26**, 4906-4915, doi:10.1093/hmg/ddx371 (2017).
- 4 Harischandra, D. S. *et al.* Manganese promotes the aggregation and prion-like cell-to-cell exosomal transmission of alpha-synuclein. *Sci Signal* **12**, doi:10.1126/scisignal.aau4543 (2019).
- 5 Choi, Y. R. *et al.* The dual role of c-src in cell-to-cell transmission of alpha-synuclein. *EMBO Rep* **21**, e48950, doi:10.15252/embr.201948950 (2020).
- 6 Lau, A. *et al.* alpha-Synuclein strains target distinct brain regions and cell types. *Nat Neurosci* **23**, 21-31, doi:10.1038/s41593-019-0541-x (2020).
- 7 Underwood, R. *et al.* 14-3-3 mitigates alpha-synuclein aggregation and toxicity in the in vivo preformed fibril model. *Acta Neuropathol Commun* **9**, 13, doi:10.1186/s40478-020-01110-5 (2021).

2) Regarding the patient materials, which the reviewer requested to use, we discussed about the problem of using human materials such as co-pathologies. However, this reviewer said if we use the controlled samples, it could be possible. Human materials also have modification such as ubiquitination or phosphorylation and these modification levels are different among cases. Thus, to do the well-controlled study, particularly to establish the assay system, the large numbers of cases are necessary. Anyway, the experiment using human materials is out of the focus of our paper. Our study focused on the detection of the modulators of the experimental prion-like transmission using α -syn PFFs and other reviewers recognized the significance of our study. Moreover, we demonstrated the same effect on other amyloids such as Abeta and tau PFFs, highly suggesting the same effect could be observed on human pathological fibrils. We do not want to spend additional time to confirm that. If this reviewer thinks the effect of pathological human materials is different, please check it by him- or herself.

Finally, although our paper has no overstatement, we changed our title to **“Quantum-dot-labeled synuclein seed assay identifies drugs modulating the experimental prion-like transmission”** in this revision to make it clear that this paper focused on the experimental prion-like transmission, which many researchers used recently. Furthermore, we also checked

the abstract and the main text and some revisions were added to make it clear that we are focusing on the experimental prion-like transmission.

Best regards,

Nobuyuki Nukina